

# Variability of temperature and ozone in the upper troposphere and lower stratosphere from multi-satellite observations and reanalysis data

Ming Shangguan[1], Wuke Wang[2,3,4], and Shuanggen Jin[5,6]

[1]School of Transportation, School of Transportation, Southeast University, Nanjing 21189, China
[2]Institute for Climate and Global Change Research, School of Atmospheric Sciences, Nanjing University, Nanjing 210023, China
[3]Joint International Research Laboratory of Atmospheric and Earth System Sciences (JirLATEST), Nanjing University, Nanjing 210023, China
[4]Collaborative Innovation Center of Climate Change, Jiangsu Province, Nanjing 210023, China
[5]Shanghai Astronomical Observatory,Chinese Acadmy of Sciences, Shanghai 200030, China
[6]School of Remote Sensing and Geomatics Engineering, Najing University of Information Science and Technology, Nanjing 210044, China

**Correspondence:** Ming Shangguan(sgming@seu.edu.cn), Wuke Wang(wuke.wang@nju.edu.cn)

**Abstract.** Temperature and ozone changes in the upper troposphere and lower stratosphere (UTLS) are important components and sensitive indicators of climate change. In this paper, variability and trends of temperature and ozone in the UTLS were investigated for the period 2002-2017 using the high quality, high vertical resolution GPS RO data, improved merged satellite data sets (SWOOSH and C3S) and reanalysis data sets (including the newest ERA5, MERRA2 and ERA-Interim). All three

reanalyses show good agreement with the GPS RO measurements in absolute values, annual cycle as well as interannual variabilities of temperature. However, relatively large biases exist for the period 2002-2006, which reveals an evident discontinuity of temperature time series in reanalyses. Based on the multiple linear regression methods, a significant warming of 0.2-0.3 K/decade is found in most areas of the troposphere with stronger increase of 0.4-0.5 K/decade in mid-latitudes of both hemispheres. In contrast, the stratospheric temperature decreases at a rate of 0.1-0.3 K/decade except that in the lower most

stratosphere (100-50 hPa) in the tropics and parts of mid-latitude in the Northern Hemisphere (NH). ERA5 shows improved quality compared with ERA-Interim and performs the best agreement with the GPS RO data for the recent trends of temperature. Similar with temperature, reanalyses ozone are also affected by the change of assimilated observations and methods. Negative trends of ozone are found in NH at 150-100 hPa while positive trends are evident in the tropical lower stratosphere. Asymmetric trends of ozone can be found for both hemispheres in the middle stratosphere, with significant ozone decrease in

NH mid-latitudes and increase of ozone in the Southern Hemisphere (SH) mid-latitudes. According to model simulations, the temperature increase in the troposphere as well as ozone decrease in the NH stratosphere could be mainly connected to the increase of Sea Surface Temperature (SST) and subsequent changes of atmospheric circulations.



# 1 Introduction

The upper troposphere and lower stratosphere (UTLS) is the key region for stratosphere-troposphere coupling and affects content of trace gases in both the troposphere and the stratosphere (Staten and Reichler, 2008; Fueglistaler et al., 2014). Temperature change in the UTLS is an important component in climate change and may act as a sensitive indicator of natural
and anthropogenic climate forcing. A net warming in the troposphere and cooling in the stratosphere were reported over the past decades, which have been attributed to the radiative impacts of increasing greenhouse gases (GHG) and changes in ozone-depleting substances (ODS) (Randel et al., 2009; IPCC, 2013). Recently, a slowing down of cooling in the lower stratosphere since 1998 (Polvani et al., 2017), or an increase of temperature in the TTL since 2001 (Wang et al., 2013) has been indicated, while the exact mechanism is still under debate (Wang et al., 2015; Polvani et al., 2017). Disagreements through different
observational data sets (Wang et al., 2012) and also between data and models (Kim et al., 2013) make it further complicated to fully understand the UTLS temperature variability and underlying mechanisms.

Measuring the temperature in the UTLS has been problematic due to its strong variability around the tropopause. Ground-based radiosonde measurements often have low temporal and spatial resolution (distributed in the northern hemisphere mostly) and also suffer from its inhomogeneity due to changes in instruments (Seidel and Randel, 2006; Wang et al., 2012). Nadir
sounding satellite measurements (e.g., Microwave Sounding Unit) can not well resolve the narrow vertical-scale features in the UTLS, which are essentially important for understanding processes related to the UTLS. Global Positioning System radio occultation (GPS RO) is a relatively new technology which measures the time delay in occulted signals from one satellite to another and provides information to derive profiles of atmospheric temperature and moisture. Since the Challenging Minisatellite Payload (CHAMP) mission launched in 2001, GPS RO has provided high quality and high vertical resolution temperature
measurements in the UTLS for almost two decades. Due to its self-calibrating and not susceptible to instrument drift (Anthes et al., 2008), the GPS RO data is well suited for long-term trend studies. Several studies have been done for detecting possible temperature trends using about 10 years of RO data in the UTLS (Ladstädter et al., 2011; Steiner et al., 2011; Wang et al., 2015; Kishore et al., 2016). A cooling ($\sim$0.88 K/decade) is observed at 30-10 hPa and a warming trend ($\sim$0.82 K/decade) is seen at 300 hPa based on the COSMIC data from July 2006 to December 2013 (Kishore et al., 2016). Due to a short time period
most of trends are not significant. One goal of this study is using the longer GPS RO record to update the recent variability of the temperature from 2002 to 2017 in the UTLS and analyze the underlying mechanisms.

Atmospheric reanalysis data, with plenty of observed data sources assimilated, are the current best estimation of the real atmosphere and provide excellent global spatial and temporal coverage of temperature. However, because of the lacking of high-quality and high-vertical-resolution temperature observations and also the low vertical resolution of the model, the re-
30 analysis data in the UTLS might be problematic (Zhao and Li, 2006; Trenberth and Smith, 2006, 2009). In addition, while assimilating many types of observations, reanalysis data suffer from instrument exchanges and perform sudden changes as new data are assimilated (Sturaro, 2003; Sterl, 2004). The second goal of this study is to validate the most recent reanalyses, including the fifth generation European Center for Medium-Range Weather Forecasts (ECMWF) atmospheric reanalysis (ERA5), the





Modern-Era Retrospective analysis for Research and Application, Version 2 (MERRA2) and the ERA-Interim (ERA-I), and assessment their performance in the UTLS region.

Ozone, which is fundamental to stratospheric and tropospheric chemistry as well as radiative budget of global climate, is closely coupled to temperature changes in the UTLS. On one hand, changes in ozone concentrations impact temperature
directly through its radiative effects (Forster et al., 2007) or indirectly through its modulation to atmospheric circulations (Polvani et al., 2017). On the other hand, most of ozone related chemical reactions are temperature dependent. Temperature changes could modify the production and loss rates and therefore impact ozone concentrations. In addition, they could both get affected by changes in atmospheric circulations. Analyzing ozone variability is then expected to be helpful for understanding processes that influences temperature changes in the UTLS.

Ozone amount in the UTLS are mainly measured by various of satellite missions. To date, the longest instrument records are Stratospheric Aerosol and Gas Experiment (SAGE)-II, which provided ozone data from 1984 to 2005 (Damadeo et al., 2013; Tummon et al., 2015). Since Aug. 2004 the Aura MLS ozone data provide continues ozone data (Waters et al., 2006). Over the past decade, there are also many new satellite-based instruments (ENVISAT, MIPAS, GOMOS, etc.) have made ozone profiles measurements but few continues data sets (Hassler et al., 2014; Tegtmeier et al., 2013). Therefore, several
institutes and projects (National Aeronautics and Space Administration (NASA), Copernicus Climate Change Service (C3S), the Stratosphere- troposphere Processes And their Role in Climate (SPARC), etc.) developed long-term vertically resolved ozone data sets for updated knowledge of long-term changes in the vertical distribution of ozone.

It is obvious that there are many different results for recent ozone variability in the UTLS regions represented by different studies. Harris et al. (2015) found some negative trend in the tropics around 15 hPa and positive trends in the lower stratosphere
at mid-latitudes and deep tropics based on the merged satellite ozone data from 1998 to 2012. Steinbrecht et al. (2017) updated the ozone profile trends for the period 2000 to 2016 and found a decreasing ozone in the tropics and at northern mid-latitude between 100 and 50 hPa. Ball et al. (2018) also indicated a continuous decline in the lower stratosphere (147-30 hPa at mid-latitudes or 100-32 hPa at tropical latitudes) from multiple satellite ozone data between 1998 and 2016. Chipperfield et al. (2018) extended the analysis to 2017 and argued that the ozone decline in the lower stratosphere is insignificant. They further
concluded that the observed variations of ozone in the LS are mainly caused by atmospheric dynamics using a 3-D chemical transport model. Therefore, whether the ozone is increase or decline recently is still under debate, while its relationship to temperature changes awaits further investigations. Another goal of this study is to analyze the recent variability of ozone in the UTLS using different merged satellite data sets.

Recently, ozone content is represented as prognostic variables in almost all current reanalysis due to its impact on strato-
spheric temperature (Dee et al., 2011; Davis et al., 2017). Although might be still problematic (Davis et al., 2017), ozone data from reanalysis has been used to detect and attribute trends in lower stratospheric ozone (Wargan et al., 2017). The ozone records from different reanalyses are also analyzed and compared to merged satellite data sets in this study. In addition, coupled chemistry climate models are widely used to attribute climate variability. NCAR's WACCM (Whole Atmosphere Community Climate Model), which is one of the two available atmospheric components of the Community Earth System Model (CESM),



is used in this study to understand underlying mechanisms that influence recent variability of temperature and ozone in the UTLS.

In this study, we investigate the seasonal-to-interannual variability and detect the recent trends of temperature in the UTLS (400-10 hPa) using the high-quality and high-vertical-resolution GPS RO data for the period 2002-2017. Recent reanalyses, especially the newest ERA5 reanalysis are also analyzed and compared to the GPS RO data. At the same time, the ozone variability in the same period from 250 to 10 hPa are compared and analyzed with the combined recorded of satellite ozone data and reanalyses. In totally, the two RO missions (COSMIC and CHAMP), two ozone merged data (The Stratospheric Water and OzOne Satellite Homogenized data set (SWOOSH) and C3S) and three reanalyses MERRA2, ERA-I and newest ERA5 are included in the study. The multiple linear regression is used to calculate the trends. The WACCM model simulations with time varying and climatological SSTs are included to check the possible influence of dynamical processing with SST for the temperature and ozone variability. In sect. 2 we provide an overview of the used observational data sets, reanalyses, model and method for trend calculation. Sect. 3 compare and analyze the temperature and ozone in absolute mean, anomalies and trend in vertically, regionally and globally. In the final section, we conclude with a summary.

## 2   Data and Methods

### 2.1   GPS RO Temperature Data

The Challenging Minisatellite Payload (CHAMP) became operational and produce 150 occultation events globally per day in 2001 (Wickert et al., 2001). Around one decade CHAMP data are available from May 2001 to Oct. 2008. In 2006 the Constellation Observing System for Meteorology, Ionosphere and Climate (COSMIC), which is a constellation of six satellites, provides more than 10 times of observations (1000-3000 occultations per day). According to previous studies (Foelsche et al., 2008; Ho et al., 2009) , the mean temperature differences between the collocated soundings COSMIC and CHAMP were within 0.1 K from 200 to 20 hPa. Many studies have demonstrated that GPS RO temperature data have good quality in the range of 8-30 km (Schmidt et al., 2005, 2010; Ho et al., 2012). Ho et al. (2009) found that results from GPS RO show a mean temperature deviation of 0.05 K with a standard deviation of 1 K in the range of 8-30 km. GPS RO data are high precision and can be used to assess the accuracy of other detection techniques such as to correct the temperature bias of radiosondes in the lower stratosphere (Ho et al., 2016). Many reanalyses have already assimilated GPS RO bending angles.

In our study, we make use of monthly mean temperature data at 400-10 hPa (approximately 6.5-30 km) for the trend analysis, with which the essential atmospheric variability can be already captured by single satellite (Pirscher et al., 2007; Foelsche et al., 2008; Ladstädter et al., 2011). More than 100 observations per month per 5 latitude grid can be provided by single satellite CHAMP. Much improved spatial coverage (more than 10 times number of profiles) appear since late 2006 due to the start of COSMIC mission. The high latitudes regions with low coverage of observations can cause large sampling errors. In consideration of large uncertainties caused by sparse data coverage at high latitudes, we consider only GPS RO data in latitude bands 60°S to 60°N here. According to the many studies (Foelsche et al., 2008; Scherllin-Pirscher et al., 2011; Ladstädter et al., 2011) the sampling errors have low effect on the trend calculation in mid-latitudes and tropics. The moisture-corrected





atmospheric temperature profile (wetPrf) products of CHAMP and COSMIC provided by the UCAR COSMIC Data Analysis and Archive Center (CDAAC) are utilized. WetPrf products using one-dimensional variational method (1DVAR) have up to 100 m vertical resolution from 0.1 to 40 km and use low resolution ECMWF ERA-I profiles as background for 1DVAR technique (Wee and Kuo, 2014). The RO data we use in this study are processed in reprocessed and post-processed categories, which can

provide stable and accurate observations for climate studies. The CHAMP wetPf2 version is 2016.2430 and COSMIC wetPrf version is 2013.3520 and 2016.1120.

Monthly zonal means on standard pressure levels (400, 350, 300, 250, 225, 200, 175, 150, 125, 100, 70, 50, 30, 20, 10 hPa) were determined, whereas 5°N non-overlapping latitude bands centered at 57.5°S-57.5°N were used. Larger discrepancies were observed for pressure levels above 400 hPa (below 6.5 km altitudes) due to high level of moisture in the lower troposphere

(Kuo et al., 2004; Kuleshov et al., 2016). Therefore we focus on the data from 400 to 10 hPa in this work. The determination of monthly zonal means were performed in four steps. Firstly, all data in a given latitude bin were averaged and standard deviation of GPS RO with 100 m interval height are calculated. Secondly, all data were re-read and data exceeding 3 times of the standard deviation from the first zonal mean were removed as outliers at 400 levels. Thirdly, GPS RO temperature profiles were interpolated to the standard pressure levels using piecewise linear interpolation and if there existed large gaps in the

profiles, no interpolation is made. In the last step the interpolated profiles are averaged to monthly mean temperatures on 17 standard pressure levels and 24 latitude bins. Monthly means with data points less than 20 observations per latitude bin are excluded for the trend analysis. Because the earliest available CHAMP data is since May 2001, we chose a comparable decadal time period from 2002 to 2017 for the temperature trend calculations.

## 2.2   Merged satellite Ozone Data

SWOOSH data set is a merged monthly mean of stratospheric ozone measurements taken by a number of limb sounding and solar occultation satellites from 1984 to present, and includes data from the SAGE-II (v7)/III(v4), UARS HALOE (v19), UARS MLS (v5/6), and Aura MLS (v4.2) instruments (Davis et al., 2016). The measurements are homogenized by applying corrections that are calculated from data taken during time periods of instrument overlap. The merged product without interpolation based on a weighted mean of the available measurements is used in this study on the pressure levels (316, 261, 215, 178, 147,

121, 100, 83, 68, 56, 46, 38, 32, 26, 22, 18, 15, 12, 10 hPa). SWOOSH uses SAGE-II as the reference for ozone data, which other ozone measurements are adjusted. After Aug. 2004 the SWOOSH merged product is essentially the v4.2 Aura MLS data. The SWOOSH data used in this work is version 2.6 in 5° latitude zones monthly means.

C3S SAGE-II/CCI/OMPS ozone products are in 10° latitude bands. The data merged 7 satellite instruments, including three instruments on board Envisate, Michelson Interferometer for Passive Atmospheric Sounding (MIPAS 2002-2012), Global

Ozone Monitoring by Occultation of Stars (GOMOS 2002-2011), SCanning Imaging Spectrometer for Atmospheric CHartographY (SCIAMACHY 2002-2012), as well as Optical Spectrograph and InfraRed Imaging System (OSIRIS 2001-), SAGE-II(1984-2005), Ozone Mapping and Profiler Suite (OMPS 2012-) and Atmospheric Chemistry Experiment Fourier Transform Spectrometer (ACE-FTS 2004-) (Sofieva et al., 2017). The absolute ozone values are adjusted to the mean of SAGE-II and OSIRIS ozone profiles in 2002-2005 (which nearly coincide also with GOMOS data). Ozone profile data are provided on an



altitude grid and ancillary information is provided with the data products to allow conversion unit. The data records combine a large number of high quality limb and occultation sensors covering a time-period suitable for trend evaluation. The evaluation of ozone trends using the merged C3S data with other data sets has been performed in (Sofieva et al., 2017; Steinbrecht et al., 2017). The results show a good agreement between C3S and other data sets and the best quality of the merged data set is in the

stratosphere in the latitude zone from $60^\circ$ S to $60^\circ$N. The altitude levels (from 10 to 50 km in steps of 1 km) are interpolated to pressure levels using linear interpolation in log-presssure space. The monthly mean ozone molar concentration are converted to volume mixing ratio using the mean temperature provided by the C3S data. The used C3S data in this work is version 3.

## 2.3   Reanalysis Data

ERA-I covers the period from 1979 until present, assimilating observational data from various satellites, buoys, radiosondes,
commercial aircraft and others (Dee et al., 2011). ERA-I includes GPS RO bending angels from CHAMP, COSMIC, GRACE, MetOp, and TerraSAR-X and satellite vertical ozones profiles from GOME/GOME-2, MIPAS, MLS, SBUV (Dee et al., 2011; Davis et al., 2017). Description of the ozone system and assessments of its qulaity have been provided by Dee et al. (2011); Dragani (2011). In this work, monthly means of ERA-I data ($2.5^\circ$x$2.5^\circ$) were averaged onto $5^\circ$ latitude bins. ERA-I reanalysis is widely used for inter-comparisons and currently used as background information for wetPrf. For these reasons, we choose it
for the comparison.

Besides ERA-I the currently newest ECMWF climate reanalyses ERA5 with the same temporal and spatial resolution is also used. ERA5 is released in 2018 by ECMWF. Compared to ERA-I, ERA5 data assimilation system uses the new version of the integrated Forecasting System (IFS Cycle 41r2) instead of IFS Cycle 31r2 by ERA-I. In addition, various newly reprocessed data sets, recent instruments and cell-pressure correction SSU, improved bias correction for radiosondes etc, are renewed in
ERA5. More information can be found in ERA5 data documentation https://copnfluence.ecmwf.int/display/CKB/ERA5+data+ documentation. Ozone and temperature monthly means at 17 standard pressure levels from 400 to 10 hPa are selected in this study.

MERRA2 is the latest atmospheric reanalysis of NASA's Global Modeling and Assimilation Office (GMAO) with data resolution $0.5^\circ$x$0.625^\circ$ (Gelaro et al., 2017). Same as ERA-I, the MERRA2 data were averaged onto $5^\circ$ latitude bins with
weighted mean method. Compared with ERA-I/ERA5, MERRA2 starts to assimilate GPS RO since Jul. 2004 and MLS ozone data since Oct. 2004 (earlier SBUV observations) (McCarty et al., 2016). For ozone data MERRA2 assimilated MLS instead of SBUV since Oct. 2004 (Gelaro et al., 2017). Monthly means of data at 15 standard pressure levels (400, 350, 300, 250, 200, 150, 100, 70, 50, 40, 30, 20, 10 hPa) are selected for the study. Wargan et al. (2017) provided a comprehensive description and validation of the MERRA-2 ozone product.

## 2.4   Model simulations

The Whole Atmosphere Community Climate Model, version 4 (WACCM4) is used here in its atmosphere-only mode. The horizontal resolution of the WACCM4 runs presented here is $1.9^\circ \times 2.5^\circ$ (latitude $\times$ longitude). More details of this model are described in Marsh et al. (2013). WACCM4 uses the finite-volume dynamical core with 66 standard vertical levels (about 1 km





vertical resolution in the UTLS). Here we use the special version with finer vertical resolution, WACCM_L103 (Gettelman and Birner, 2007), with 103 vertical levels and about 300 m vertical resolution in the UTLS. This high vertical resolution version has been proved for a better representation to the detailed thermal structure as well as interannual-to-decadal variations in the UTLS (Wang et al., 2013, 2015).

A hindcast simulation (hereafter termed as the Transient run) was done for the period 1995-2017 to reproduce the recent temperature and ozone variability in the UTLS. The model was forced by observed Greenhouse Gases (GHGs), ozone depleting substances (ODSs) and solar irradiances, nudged QBO (Quasi-Biennial Oscillation) (Matthes et al., 2010) and prescribed SSTs (using the HadISST data set (Rayner et al., 2003)). The first 7 years (1995-2001) are not analyzed for a spin-up. Based on this simulation, a FixSST run was integrated for the same period and using the same climate forcing except that SSTs were fixed

to climatological values. The difference between these two simulations indicate SST impacts on the atmosphere.

## 2.5   Trends Methodology

From the monthly zonal mean time series the seasonal cycle is firstly calculated, and monthly zonal anomalies are estimated by subtracting the seasonal cycle from each individual monthly mean. This data analysis was performed for each data set and zonal bin. The calculated anomalies are the basis for trend calculations. The QBO and ENSO (El-Niño Southern Oscillation) are the

15 most important phenomenons that affects interannual variability of the UTLS. To exclude the effects of QBO and ENSO, we apply a simple multiple linear regression (MLR) based on the temperature monthly anomalies (Eq. 1) (von Storch and Zwiers, 2002).

$$y(t) = a_0 + a_1 \cdot t + a_2 \cdot ENSO(t) + a_3 \cdot QBO50(t) + a_4 \cdot QBO30(t) \qquad (1)$$

The regression coefficients comprise a constant $a_0$, the trend coefficient $a_1$, the ENSO coefficient $a_2$, the QBO coefficient

$a_3$ and $a_4$ and the solar cycle coefficient $a_4$. The QBO30 and QBO50 indexes for the period 2002-2017 are normalized to unit variance from the CDAS Reanalysis data, which are the zonally averaged winds at 30 and 50 hPa and taken from over the equator (http://www.cpc.ncep.noaa.gov/data/indices/). The ENSO MEI indexes are obtained from NOAA on the six main observed variable (sea-level pressure, zonal and meridional components of the surface wind, sea surface temperature, surface air temperature and total cloudiness fraction of the sky) over the tropical Pacific (http://www.cdc.noaa.gov/people/klaus.wolter/

MEI/table.html). The t-statistic is used to test for a significant linear regression relationship between the response variable and the predictor variables. The significance level is set to be 0.05.

## 3   Results and Analysis

### 3.1   Time series of temperature

Figure 1 shows the initial time series of zonal mean temperature at 400 hPa from the GPS RO observations and different

reanalyses (ERA5, MERRA2 and ERA-I) as well as the relative differences between reanalyses and the GPS RO data. Three latitude bands are selected to indicate temperature variations in the tropics (10°S-10°N), mid-latitude in the NH (25°N-45°N)





and SH (25°S-45°S). Seasonal variations are relatively small in the tropics while evident annual cycle can be seen in mid-latitudes of both hemispheres. Generally, reanalyses show good agreement with the GPS RO in monthly absolute values as well as seasonal variations. Seen from the differences between reanalyses and the GPS RO, the bias of ERA5 and ERA-I are less than 0.3 K except in mid-latitude for the period 2002-2006, which shows bias up to 0.6 K. As the 5th generation of the

5 ECMWF reanalysis, ERA5 shows slightly better agreement than ERA-I in the tropics. Temperature in ERA-I is obviously warmer than the GPS RO of about 0.1-0.2 K, while ERA5 temperature shows differences of less than 0.1 K compared with the GPS RO data. Warm bias of 0.3 K is seen for MERRA2 in all selected regions, which is over 0.9 K in mid-latitude for the period 2002-2006.

At 100 hPa, as indicated by Figure 2, more evident seasonal variations of temperature can be seen in the tropics, with similar

amplitude to that in mid-latitudes of both hemispheres. It is note worthy that the annual cycle of temperature at 100 hPa in mid-latitudes are more disturbed than at 400 hPa in the upper troposphere. Compared with the GPS RO temperature, ERA-I shows evident cold bias in the tropics during the period 2002-2006. For ERA5, such bias is largely corrected. For the later period 2007-2017, the differences between three reanalyses and the GPS RO are comparable in magnitude, although the ERA5 shows slightly better agreement with GPS RO measurements. In mid-latitudes of both hemispheres, very similar characteristics

can be seen through the three reanalyses, which show slightly better agreement with the GPS RO than in the tropics. However, relatively large bias can still be seen in the early stage from 2002 to 2006.

Temperature in the lower stratosphere (70 hPa) shows clear annual cycle in the tropics (figure 3(a)). However, the annual minimum and maximum values vary year-to-year, which indicate influences from the QBO. Large sub-seasonal fluctuations of temperature can be seen in mid-latitude of the NH, which is obviously different from that in the SH. That is related to

20 strong equatorial as well as extra-tropical wave activities in this region. Again, large differences up to 1 K exists between the reanalyses and the GPS RO observations during the first stage 2002-2006. ERA5 shows obvious cold bias in all selected regions while MERRA2 is anomalously warmer than the GPS RO in mid-latitudes of both hemispheres. ERA-I, however, has no consistent warm or cold bias and shows the best agreement with the GPS RO for the period 2002-2006. For the latter stage of 2007-2017, ERA5 shows the best agreement with observations (differences within 0.2 K) while the other two reanalyses are

25 slightly (about 0.2 K) warm biased.

Note that the bias is particularly large during 2001-2006 for all reanalyses. This should be related to the assimilation of large number of COSMIC data since late 2006, which may cause sudden changes in reanalyses (Sturaro, 2003; Sterl, 2004). At the same time, the GPS RO data could be also affected by the transition from the single CHAMP satellite to six COSMIC satellites since late 2006. To quantify the sampling errors and bias between two RO missions, we compared COSMIC and

30 CHAMP monthly means for their overlap period of Jun. 2006-Sep. 2008. Figure 4(b) shows that COSMIC monthly zonal mean temperatures are consistent colder (0.1-0.2 K) than CHAMP in the stratosphere. This is consistent with previous studies (Foelsche et al., 2008; Ho et al., 2009), although the differences between CHAMP and COSMIC are slightly larger than 0.1 K in some areas in the middle stratosphere (50-10 hPa). According to Schröder et al. (2007); Leroy et al. (2018) the cold bias between CHAMP and COSMIC is the consequence of a change in the signal-to-noise ratio from 550 in CHAMP to 700 in

COSMIC. In addition the ribbed pattern in the meridional structure of the bias in the figure 4 is a consequence of sampling



error (Leroy et al., 2018). The bias between COSMIC and CHAMP was computed from the 28-month period of overlap and removed from CHAMP-retrieved temperature for the further analysis in this work.

Figure 5 shows differences between three reanalyses and the corrected CHAMP for the period 2002-2006 and COSMIC for the period 2007-2017, respectively. For the first stage, MERRA2 shows warm bias of 0.1-0.3 K in the upper troposphere,

cold bias of 0.1-0.4 K in the lower stratosphere and warm bias of 0.1-0.5 K in the tropical middle stratosphere. ERA5 shows relatively small cold bias of 0.1-0.2 K for almost the whole UTLS region. ERA-I shows warm bias of 0.1-0.5 K around the tropical tropopause and cold bias of 0.1-0.5 K in the middle stratosphere in both tropics and SH. For the second stage, differences between all three reanalysis and the GPS RO are much smaller. That is because the reanalyses are better constrained by large number of COSMIC measurements. MERRA2 shows cold bias of about 0.1 K in the upper troposphere, warm bias

of 0.1 K in the lower stratosphere and cold bias of 0.1 K in the middle stratosphere. ERA5 shows perfect agreement to the COSMIC with differences less than 0.1 K in most of the UTLS region except that in northern mid-latitudes (100-50 hPa) with warm bias 0.1 K. Bias in ERA-I is also quite small with warm bias of about 0.1 K only in the tropics around the tropopause and southern mid-latitudes near 10 hPa.

In summary, reanalyses show very good agreement with the GPS RO measurements in sub-seasonal to seasonal variations of

15 temperature in the UTLS region. For the climatological values, a notable change around late 2006 can be found in all reanalyses. Relatively large bias of 0.1-0.5 K can be seen in MERRA2 and ERA-I for the first stage 2002-2006 while very good agreement can be seen between all reanalyses and the GPS RO measurements for the 2007-2017. As the newest reanalysis, ERA5 shows relatively small bias of 0.1-0.3 K during 2002-2006 and performs the best agreement with GPS RO in general.

### 3.2  Interannual Variability of temperature

Figure 6 shows one example of deseasonalized monthly anomalies of temperature in the tropical upper troposphere (10°S-10°N, at 150 hPa). As demonstrated in Figure 6a, temperature performs clear interannual variations, which is related to ENSO and QBO as indicated by previous studies (Randel and Wu, 2015). While the period of analysis is relatively short, such interannual fluctuations may significantly affect the calculation of linear trends. To estimate the influences of ENSO and QBO, a multiple linear regression method is applied as introduced in section 3.1. Figures 6d-f indicate contributions of QBO50,

QBO30 and ENSO, respectively. ENSO contributes the largest and significant interannual variations of temperature in tropical upper troposphere with amplitude of about 0.5 K while QBO has only small and insignificant contributions. At lower levels in the free troposphere, the QBO contribution is getting less and the impacts of ENSO are more significant. Reanalyses perform a very good agreement with each other as well as the GPS RO in ENSO related contributions (Figure 6f) but show larger spread for QBO contributions. By such a multiple linear regression, the influences of ENSO and QBO are expected to be excluded

and the linear trend is therefore estimated. Seen from Figure 6c, GPS RO indicates an increase of 0.4 K in temperature for the whole period 2002-2017. MEERA2 shows a stronger increase of 0.6 K while the ERA-I is almost flat. ERA5, however, shows the best agreement with GPS RO with an increase of about 0.5 K.

In the lower stratosphere, as illustrated in Figure 7, interannual variations of temperature are dominated by QBO, with amplitudes of over 1 K for QBO50. The ENSO effects are insignificant with an amplitude of about 0.5 K. GPS RO indicates



an increase of 0.5 K as seen in Figure 7c. MERRA2 and ERA-I/ERA5 show similar increase of 1 K which is stronger than GPS RO. The relative contributions of ENSO and QBO to interannual variations of zonal mean temperatures in the UTLS are shown in Figures 8-10.

Consistent with previous studies (Randel and Wu, 2015), ENSO is associated with warm temperature anomalies of 0.1-0.4 K in tropical upper troposphere and cold temperature anomalies of 0.1-0.4 K above the tropopause in the tropics (Figure 8). Contrast to the tropics, anomalous cold temperatures can be seen in the sub-tropics below 100 hPa while warm temperature anomalies exist above 100 hPa. All three reanalyses show consistent pattern as seen in GPS RO associated with ENSO. However, ENSO signals in tropical upper troposphere are slightly stronger in MERRA2 compared with GPS RO and other reanalyses, while ERA5 shows relative weak signals in subtropics above 100 hPa.

As a stratospheric phenomenon, QBO affects the temperature mainly in the upper atmosphere above 100 hPa. The spatial structure of temperature anomalies associated with terms of QBO50 and QBO30 are shown in Figures 9-10. QBO50 is associated with warming in the lower most stratosphere (100-50 hPa) and cooling in middle stratosphere (50-15 hPa) in the tropics. Sub-tropics and mid-latitudes, however, show out-of-phase variations with significant warming signals. QBO30 contributes to similar temperature variations except that the signals are spatially orthogonal with the patterns associated with QBO50 (Figure 10). Reanalyses show very good agreement with GPS RO in both spatial pattern and magnitude for QBO related temperature variations as illustrated in Figures 9-10.

### 3.3 Linear trend of temperature

Figure 11 summarize the spacial distribution of temperature trends based on GPS RO and reanalyses for the time period 2002-2017. From the GPS RO measurements, positive trends of 0.2-0.3 K/decade are significant in most areas of the troposphere with stronger warming up to 0.4-0.5 K/decade in mid-latitudes of both hemispheres. At the same time, negative trends of 0.1-0.3 K/decade are evident in the stratosphere except that in the lower most stratosphere (100-50 hPa) in the tropics and parts of mid-latitude in the NH, whereas the temperature trends are positive. Reanalysis data show good agreement with the GPS RO for the general pattern of temperature trends. However, neutral trends are found in MERRA2 in the tropical free troposphere (400-200 hPa), which could be related to the observed warm bias during 2002-2006 in MERRA2 as illustrated in Figure 5. ERA-I shows insignificant negative trends around 225-175 hPa in the tropics (20°S-20°N), which is not observed by other data sets. According to Simmons et al. (2014), local degradation occurs near the sub-tropical tropopause whereas substantial amounts of warm-biased aircraft data are assimilated since 1999. After 2006, while large number of COSMIC data is assimilated, this warm bias disappeared. This anomalous warm temperature for the short period 1999-2005 leads less warming in this region as estemated by ERA-I. Such bias has been corrected in ERA5. Very good agreement can be seen between ERA5 and the GPS RO with very similar spacial pattern and comparable magnitude of warm in the troposphere.

In the stratosphere, the negative trends in MEERA2 are too strong while that in ERA-I are too weak and less significant in the SH. At the same time, positive trends in NH are stronger in both ERA5 and ERA-I than that in GPS RO. Again, ERA5 shows the best agreement with GPS RO measurements with consistent pattern and comparable magnitude except that the negative trends in mid-latitude (around 30°N) lower stratosphere (150-50 hPa) in ERA5 are weaker and less significant than that in GPS



RO. At 10 hPa the all data sets show negative trends except ERA-I. According to Simmons et al. (2014), the large differences between MERRA2 and ERA-I at 10 hPa are associated with differing treatments of the change from SSU to AMSU-A and the availability of increasing amounts of largely unadjusted radiosonde data. While cell-pressure correction to SSU has been done in ERA5, ERA5 data show similar cooling trends to observations at 10 hPa. Also notable difference between GPS RO

and reanalyses can be seen in the tropics (5°S-20°N) around the tropopause. Neutral or insignificant positive trends are found by GPS RO in this region while ERA5 and ERA-I show significant positive trends (0.4 K/decade). This is related to the cold bias of ERA5 in this region during 2002-2006. In addition, as a transition zone between the troposphere and the stratosphere, opposite sign could appear in neighboring layers below or above the tropopause, which causes large uncertainties in estimated trends around the tropopause.

Overall, the ERA5 data show the best agreement with the GPS RO measurements among most of areas as demonstrated in this study, which could also be confirmed by table 1. Table 1 shows the temperature trends based on GPS RO and reanalysis data sets in three regions (SH, NH, TP) at selected pressure levels (250, 150, 70, 50, 20, 10 hPa). To explain the underlying mechanisms of the illustrated temperature trends, two WACCM simulations as described in section 2.4 were employed. Figure 12 shows the temperature trends from the Transient run and the FixSST run as well as their differences. The Transient run

with varying SST (Figure 12a) shows comparable positive trends (0.2-0.3 K/decade) in the troposphere and negative trends (0.1-0.5 K/decade) in the stratosphere (see Figure 11 for a comparison). While the SSTs are fixed to climatological values, which means only radiative effects from GHGs and ODSs are included, the positive trends in the troposphere disappear or becomes much weaker (Figure 12b). This reveals that dynamic processes are the main reason for the warming temperature trends in troposphere, which can be confirmed by the differences between these two runs (Figure 12c). The positive trends

above the tropical tropopause (100-50 hPa) as well as negative temperature trends in the stratosphere (tropics and SH) persist in the FixSST run. This indicates that such changes in temperature are dominated by radiative effects associated with increases of GHGs and ozone recovery. The significant cooling trends at 150-50 hPa in the NH subtropics and insignificant trends above are connected with both radiative and dynamical effects.

### 3.4   Coupling with ozone

As described in the Introduction, changes in temperature and ozone are closely coupled to each other. Analyzing ozone variations at the same time is therefore useful for attributing temperature trends in the UTLS. Figure 13 shows the initial ozone time series from the SWOOSH, C3S, MERRA2 and ERA5 as well as their differences using the SWOOSH data as a reference in three regions at 70 hPa. The ERA-I is not included here for ozone analysis because it does not assimilate so much ozone measurements as ERA5 and MERRA2. Although the phase and amplitude agree well in general, the absolute ozone values

have large differences between different data sets. Obvious missing data and extreme values exist in both SWOOSH and C3S data sets during 2002-2004, while a discontinuity in the MERRA2 and ERA5 time sereis occurs in mid-2004 when Aura MLS mission starts. As illustrated in Figure 13, extreme large values are observed by SWOOSH and C3S around 2003. The reason is the limited number of observation in this period, which could cause large sampling errors and uncertainties in ozone data. At the same time, since the reanalysis is less constrained during this period, large bias can be seen in both MERRA2 and ERA5




compared to observations (Figures 13b, d and f). After 2006, SWOOSH uses MLS ozone data only (Davis et al., 2016) and MERRA2 also uses MLS instead of SBUV ozone data since Oct. 2004 (Gelaro et al., 2017). Therefore the MERRA2 ozone data have good agreement with SWOOSH data. Another discontinuity in the MERRA2 and ERA5 time series occurs around 2015. According to McCarty et al. (2016), MERRA2 starts to use the version 4.2 MLS ozone data instead of version 2.2 since

June 2015, which cause data discontinuities at 250-70 hPa. As seen in Figures 13b, d and f, ozone in MERRA2 is 50-150 ppbv lower than that in SWOOSH and C3S. ERA5 combined more satellite data (SBUV and MLS) than MERRA2, which leads to larger variability of ozone in ERA5 since the different data sets and different ways for merging the data have large influences on the ozone data. The missing data and extreme values in SWOOSH and C3S, as well as the data discontinuities in MERRA2 and ERA5 around years 2004 and 2015 can also be seen at other pressure levels (See Figures S1-S2 for details).

To examine the connection between the vertical temperature changes and ozone distribution, ozone trends are analyzed in the stratosphere from 250 to 10 hPa. In consideration of poor ozone data quality during 2002-2004, the trends are calculated for the period 2005-2017 using the MLR method same as for temperature (14). SWOOSH and C3S ozone trends show good agreement in spacial distribution as well as magnitude in general (Figures 14a and b). From 250 to 100 hPa, ozone trends are mainly insignificant or opposite in sign by different data sets due to the large uncertainties of ozone data in this region.

Asymmetry trends in two hemispheres, with significant decrease of ozone in NH mid-latitudes at 100-10 hPa and increase of ozone in SH mid-latitudes are found at 50-10 hPa. At 100-50 hPa, ozone is decreasing in NH mid-latitudes, which is also found by Steinbrecht et al. (2017) based on various ozone data sets of 2000-2016.

Around 70 hPa, unrealistic negative trends are found by MERRA2 data (Figure 14c), which are related with negative MERRA2 bias during 2015-2017. In contrast, stronger positive trends in tropics and SH mid-latitudes and less negative trends

in NH mid-latitudes are found by ERA5 data from 50 to 20 hPa, which is consequence of positive ERA5 bias during 2015-2017 in these regions (Figures S1-S2). If the ozone trend is calculated for the period 2002-2017, the SWOOSH and C3S show similar spatial pattern (Figure S3) as seen in Figure 14. However, the MERRA2 and ERA5 show very different results (Figure S3c-d) as that in Figure 14.

Figure 15 shows the ozone trends from two model simulations as well as their differences. The ozone trends based on the

model simulation with varying SST show similar trends as SWOOSH and C3S data. Insignificant trends are found at 200-100 hPa in most regions. The maximum negative trends (-150 ppbv/decade) located around 30 hPa in NH mid-latitudes while the maximum positive trends at 10 hPa around 20°S. While the SSTs are fixed to climatological values, ozone increases from the tropics to SH mid-latitudes in the middle stratosphere (30-10 hPa) and negative trends in the NH mid-latitudes from 100 to 10 hPa become much weaker (Figure 15b). The differences between these two runs, which indicate contributions from

SSTs, show similar spacial pattern with the Transient run as well as observations. This confirms that dynamic processes are dominated for ozone trends in the middle stratosphere (100-10 hPa in NH and 30-10 hPa in tropics and SH). For the tropical lower stratosphere (20°S-20°N, 50-30 hPa), ozone trends are determined by a combination of ODSs and SSTs Figures 15b-c.

Considering the coupling between changes in ozone and temperature, the tropospheric warming and decreases of ozone are related to SST changes and subsequent modulation of atmospheric circulation. As seen in Figure S4 SSTs are significantly in-

creased during 2002-2017 almost globally except in the North Atlantic and the Southern Ocean. Such increase in SSTs would





warm up the atmosphere and lead to strengthening in upward motion of the atmosphere, which lifted more poor ozone lower tropospheric air to the upper troposphere and reduced ozone concentrations there. The enhanced upward motion could lead to cooling in temperature and less ozone in tropical lower stratosphere. However, that is not the truth as seen in observations with temperature warming and ozone increase in that region (Figures 11 and 14). This ozone increase should be partly related to

5 the reduced emissions of ODSs since the Montreal Protocol, which contributes to the temperature warming in tropical lower stratosphere due to its radiative effects. The decreases in ODSs also partly lead to ozone increases in the middle stratosphere (Figure 14b). Note that the SST increases are asymmetry in the two hemispheres. SST increases are stronger and more significant in the NH than that in the SH. This leads to asymmetric changes of atmospheric circulation, e.g. the Brewer Dobson Circulation (BDC) in the stratosphere, and contribute to asymmetric distributions of temperature and ozone trends in the middle

stratosphere (Figures 11 and 14).

## 4 Conclusions and Discussion

The recent variability and trends of temperature in the UTLS have been studied for the period 2002-2017 using the high quality, high vertical resolution GPS RO data. The newest ERA5 reanalysis, as well as the MERRA2 and the ERA-I reanalyses are evaluated for seasonal-to-interannual variations as well as linear trends of temperature in the UTLS. While temperature

is closely coupled with ozone, UTLS ozone from new and improved satellite data sets (SWOOSH and C3S) as well as the reanalyses (ERA5 and MERRA2) is analyzed to attribute recent temperature changes.

  In general, all three reanalyses show good agreement with the GPS RO measurements for the annual cycle of temperature with consistent phase and comparable amplitude. However, relative large biases can be seen between reanalysis data set and GPS RO for the period 2001-2006, which reveals an evident discontinuity of temperature time series in reanalyses. That is

20 caused by the lack of observations and less constrained reanalysis in the first stage and large amounts of data from the COSMIC satellite mission since 2007. Such discontinuity in reanalysis should be carefully considered while using the reanalysis data analyzing trends. ERA5, as the newest generation of reanalysis from ECMWF, show obvious improvement refers to ERA-I and best agreement with GPS RO measurements.

  Temperature in the UTLS performs significant interannual variations which has been well known that related to ENSO and

25 QBO. Based on a multiple linear method, the relative contributions of ENSO and a pair of orthogonal time series of QBO (QBO50 and QBO30) are estimated from the GPS RO measurements as well as reanalysis data sets. Signals of ENSO and QBO show very good agreement between all three reanalyses and the GPS RO data, which indicates that the reanalyses are able to capture interannual variations of temperature in the UTLS.

  Nearly 2 decades of temperature data were analyzed by a MLR method to determine trends in the UTLS. A significant

warming of 0.2-0.3 K/decade can be seen in most areas of the troposphere with stronger increase of 0.4-0.5 K/decade in mid-latitudes of both hemispheres. Contrast to the troposphere, the stratospheric temperature decreases at a rate of 0.1-0.3 K/decade except that in the lower most stratosphere (100-50 hPa) in the tropics and parts of mid-latitude in the NH, whereas the temperature trends are positive. Again, ERA5 shows improved quality compare with ERA-I and performs the best




resemblance with the GPS RO data while insignificant warming trends exist in the tropical troposphere and too strong cooling can be seen in MERRA2.

In the stratosphere, ozone distribution is highly correlated with the temperature change. Similar with temperature data, reanalysis ozone are affected by the change of assimilated observations and methods. Negative trends of ozone are dominated

in the NH at 150-100 hPa. In the tropical lower stratosphere, increases of ozone are evident. Asymmetric trends of ozone can be found for two hemispheres in the middle stratosphere, with significant ozone decrease in NH mid-latitudes and increases of ozone in SH mid-latitudes. Around the tropopause, trends are small and large differences between data sets are found. Further study and longer time series are needed for trend analyses in these regions. Overall, large biases exist in reanalysis and it is still challenging to do trend analysis based on reanalysis ozone data.

According to model simulations, the temperature increase in the troposphere as well as ozone decrease in the NH stratosphere could be mainly connected to the increase of SST and subsequent changes of atmospheric circulations. Ozone increases around 50 hPa, decreases around 30 hPa and increases from 20 to 10 hPa in the tropics are also closely related to SST changes. This support the results of Chipperfield et al. (2018) which concluded that dynamical changes play an important role for the ozone variability in the UTLS. Ozone increases in the tropical lower stratosphere are also related to the reduced ODSs emissions since

the Montreal Protocol. This increased ozone contributes to a temperature increase in this region due to the radiative effects of ozone. Because of the decrease of ODSs emissions, stratospheric ozone seems to be recovery in the tropics as well as in SH. However, temperatures are decreased in this region. While the SST changes could not explain this neither, such stratospheric cooling could be related to GHGs increases and subsequent radiative cooling.

Recent temperature and ozone trends have been calculated by a MLR method based on observational data sets. However,

trend assessments over short period of 1-2 decades are largely uncertain since the calculated trends are sensitive to start or end date (Santer et al., 2017). As RO data are acquired over longer periods with large number of observations (more than 10000 per day) by COSMIC2, the climate signal will emerge robustly and be more reliable for the temperature trends and variability studies in the UTLS.

*Author contributions.* M. Shangguan performed the computational implementation and the analysis, created the figures, and wrote the first

draft of the paper. All authors contributed to the study design. W. Wang made the model simulations and provided advice on the analysis design and contributed to the text. S. Jin contributed to the text.

*Acknowledgements.* This work was supported jointly by the Natural Science Foundation of Jiangsu Province (BK20170665), the National Natural Science Foundation of China (41705023) and the Postdoctoral Science Foundation of China (2017M610319).The numerical calculations in this paper have been done on the computing facilities in the High Performance Computing Center (HPCC) of Nanjing University.

We thank CDAAC for the use of the COSMIC GPS-RO data sets, the NOAA for QBO and MEI data, the ECMWF for the ERA-Interim and ERA5 data, the NASA GSFC for MERRA2 data, the NOAA Chemical Sciences Division for the SWOOSH data and Copernicus Climate Change Service for SAGE-II/CCI/OMPS ozone products.



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



**Table 1.** Temperature trends in K/decade per Decade during different regions (SM: 25°S-45°S; NM: 25°N-45°N; TP: 10°S-10°N) from 250 to 10 hPa for the period 2002-2017,* marked significant at 5% level.

| Levels [hPa] | 250 | 150 | 70 | 50 | 20 | 10 |
|---|---|---|---|---|---|---|
| $SM_{GPSRO}$ | 0.31±0.13* | 0.41±0.16* | 0.22±0.19* | -0.06±0.2 | -0.38±0.25* | -0.53±0.33* |
| $SM_{MERRA2}$ | -0.03±0.13 | 0.44±0.16* | 0.29±0.19* | 0±0.2 | -0.39±0.25* | -1.16±0.34* |
| $SM_{ERA5}$ | 0.36±0.13* | 0.55±0.16* | 0.43±0.19* | 0.12±0.19 | -0.32±0.25* | -0.63±0.33* |
| $SM_{L103}$ | 0.14±0.14* | 0.26±0.21* | 0.09±0.2 | -0.04±0.25 | -0.32±0.23* | -0.6±0.26* |
| $NM_{GPSRO}$ | 0.41±0.15* | 0.18±0.15* | -0.23±0.35 | -0.25±0.2* | -0.08±0.31 | -0.19±0.36 |
| $NM_{MERRA2}$ | 0.19±0.13* | 0.15±0.15* | -0.16±0.23 | -0.02±0.24 | -0.11±0.29 | -0.76±0.33* |
| $NM_{ERA5}$ | 0.4±0.13* | 0.26±0.15* | -0.08±0.22 | 0.03±0.25 | -0.02±0.3 | -0.26±0.34 |
| $NM_{L103}$ | 0.25±0.14* | -0.15±0.19 | -0.39±0.39* | -0.24±0.29 | -0.2±0.35 | 0.09±0.38 |
| $TP_{GPSRO}$ | 0.3±0.11* | 0.26±0.1* | 0.32±0.29* | 0.16±0.32 | -0.4±0.36* | -0.27±0.38 |
| $TP_{MERRA2}$ | 0.03±0.11 | 0.39±0.1* | 0.63±0.31* | 0.18±0.3 | -0.26±0.36 | -0.99±0.35* |
| $TP_{ERA5}$ | 0.27±0.11* | 0.31±0.1* | 0.61±0.3* | 0.28±0.3 | -0.21±0.35 | -0.36±0.36* |
| $TP_{L103}$ | 0.22±0.12* | 0.12±0.14 | 0.28±0.44 | 0.11±0.33 | -0.47±0.48 | -0.45±0.36* |





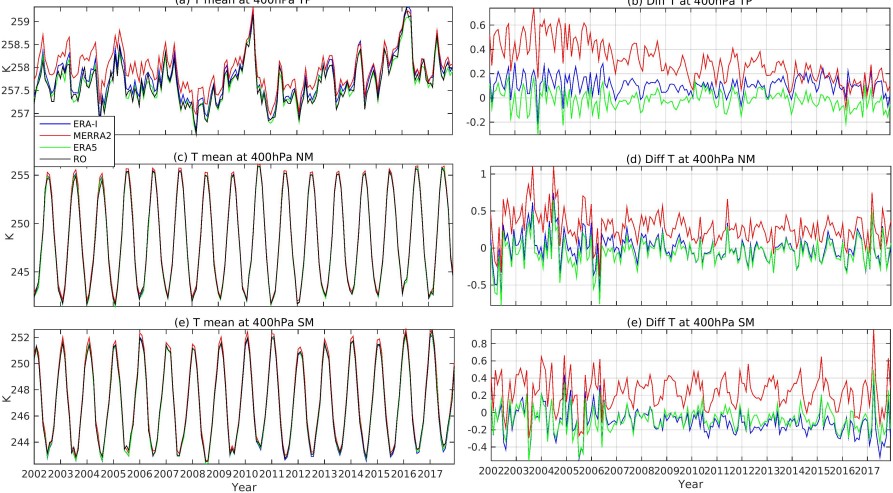

**Figure 1.** Left monthly mean temperature in K at pressure level 400 hPa through three latitude bands of TP(10°S-10°N)(a), NM(25°N-45°N) (c), NM(25°S-45°S) (e); Right corresponding differences between tree renanalyses and the GPS RO in figures (b), (d) and (e); Model with 103 levels (margin), ERA5 (green), ERA-I (blue), MERRA2 (red) and GPS RO (black) are included.

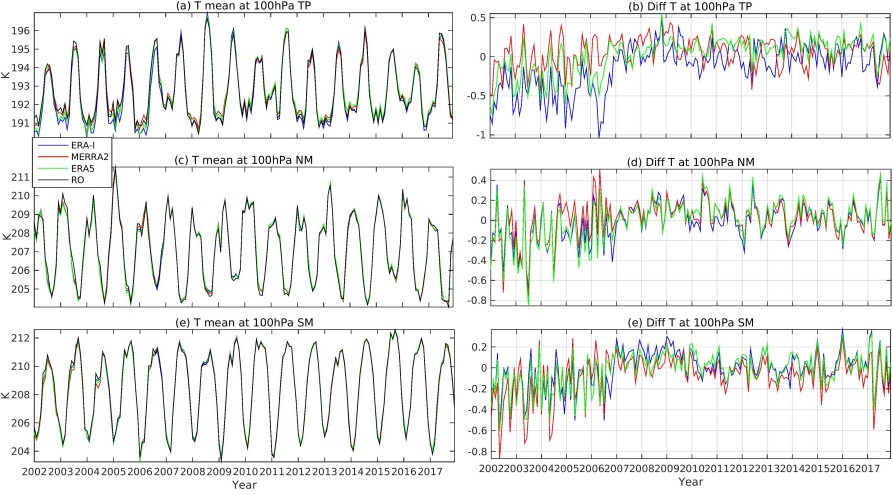

**Figure 2.** Same as figure 1 only for 100 hPa.

(





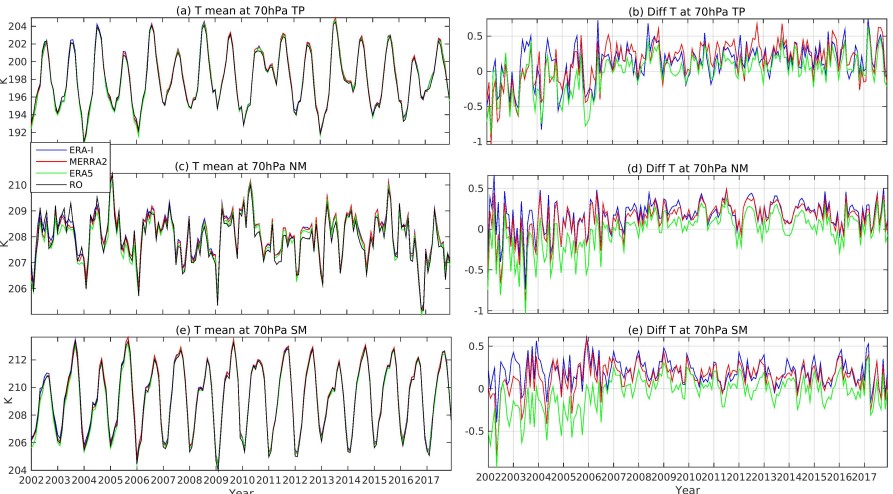

**Figure 3.** Same as figure 1 only for 70 hPa.

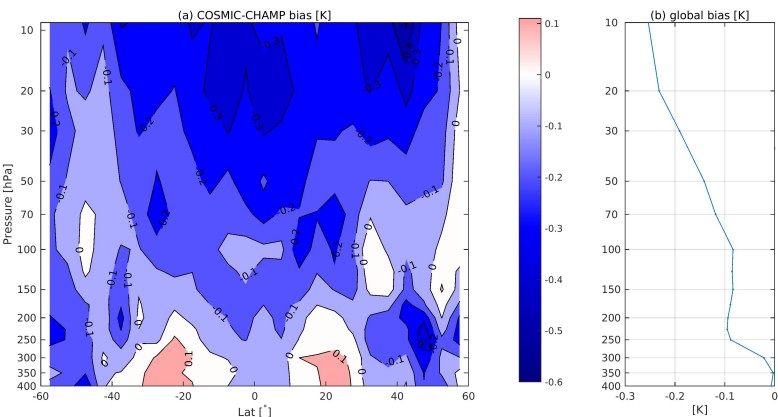

**Figure 4.** The bias in temperature climatology as retrieved from CHAMP and COSMIC RO data. The two mission obtained data during a 28 month overlap period from Jun. 2006 to Sep 2008. (a) The difference of monthly zonal mean temperature; (b) The corresponding averaged difference for each layer.





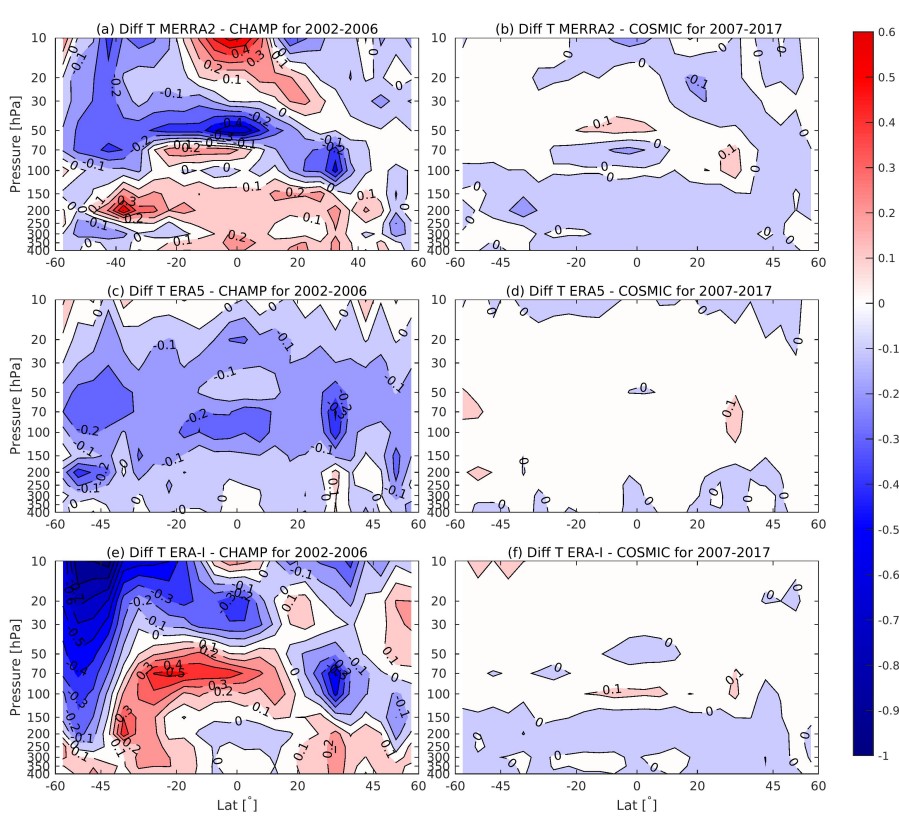

**Figure 5.** Differences of mean temperature between three reanalyses and CHAMP from 400 to 10 hPa for 2002-2006 (a, c and e) and between three renalayses and COSMIC for 2007-2017 (b, d and f).





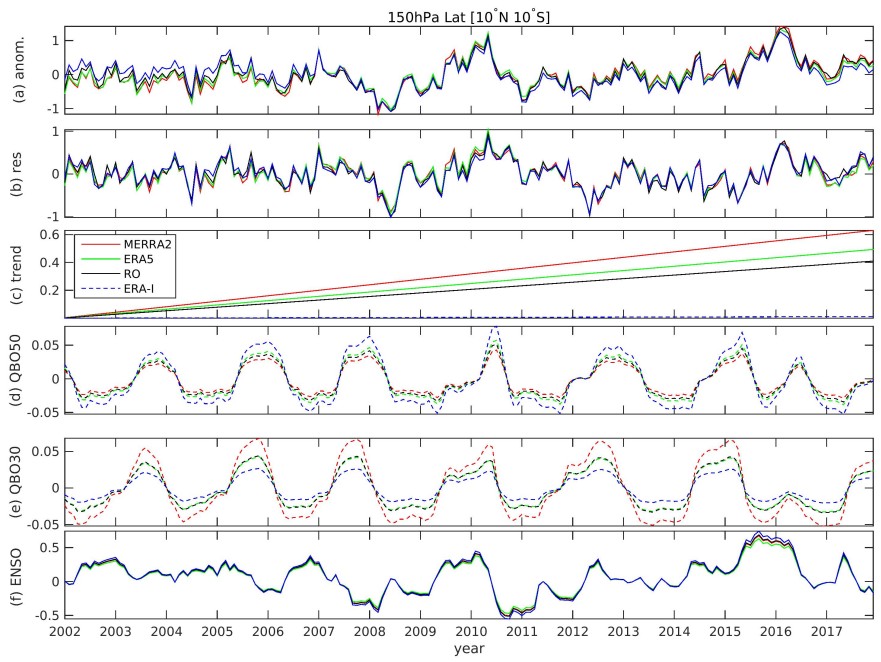

**Figure 6.** Temperature anomaly at pressure level 150 hPa in the tropics(10°S-10°N) from ERA5 (green), ERA-I (blue), MERRA2 (red) and GPS RO (black) (a); (b) The corresponding residual; (c) The linear terms; (d) The QBO50 terms; (e) The QBO30 terms and (f) the ENSO terms; The solid lines in (c-f) marked the siginificant terms and the dash lines in (c-f) marked the insiginifcant terms.





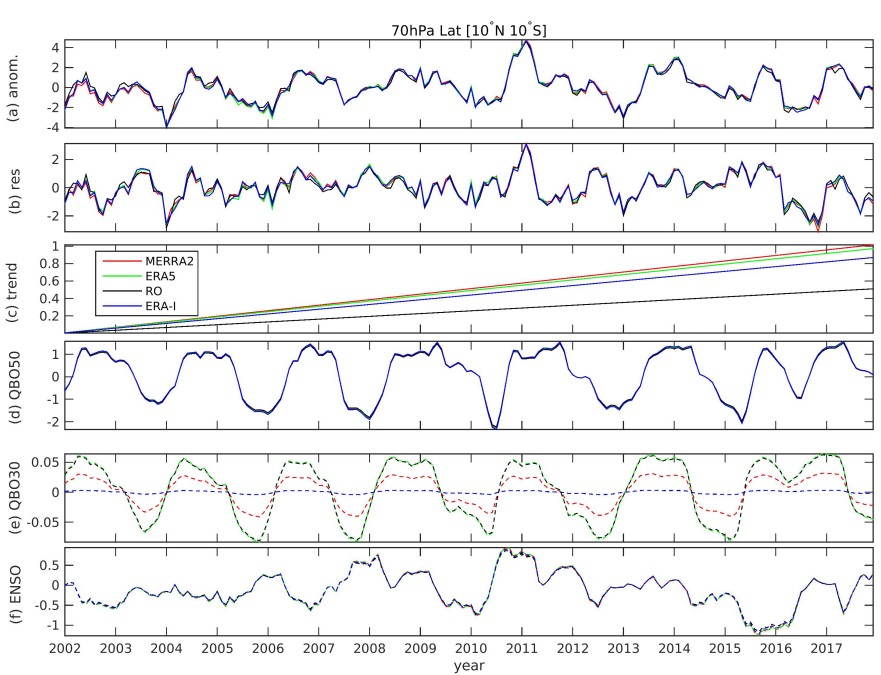

**Figure 7.** Same as figure 6 only for 70 hPa.





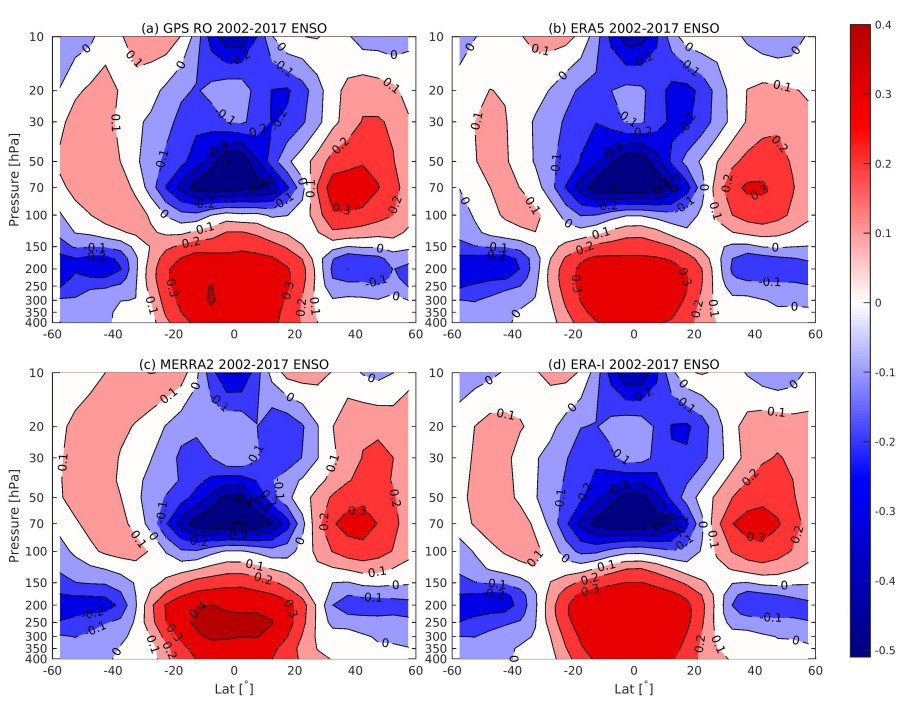

**Figure 8.** ENSO related temperature anomalies of GPS RO (a), ERA5 (b), MERRA2 (c) and ERA-I (d).





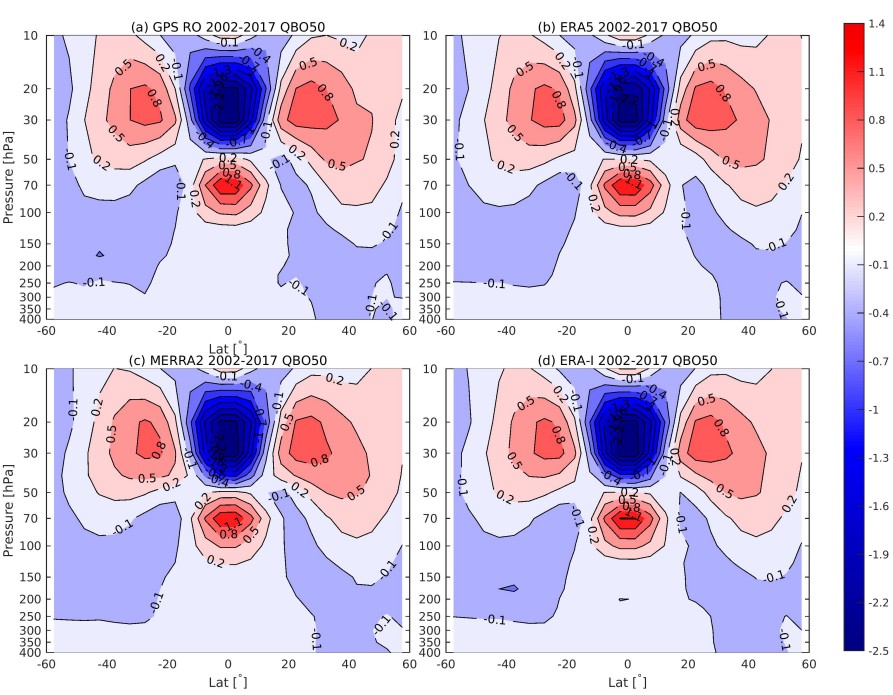

**Figure 9.** Same as figure 8 but for QBO50.





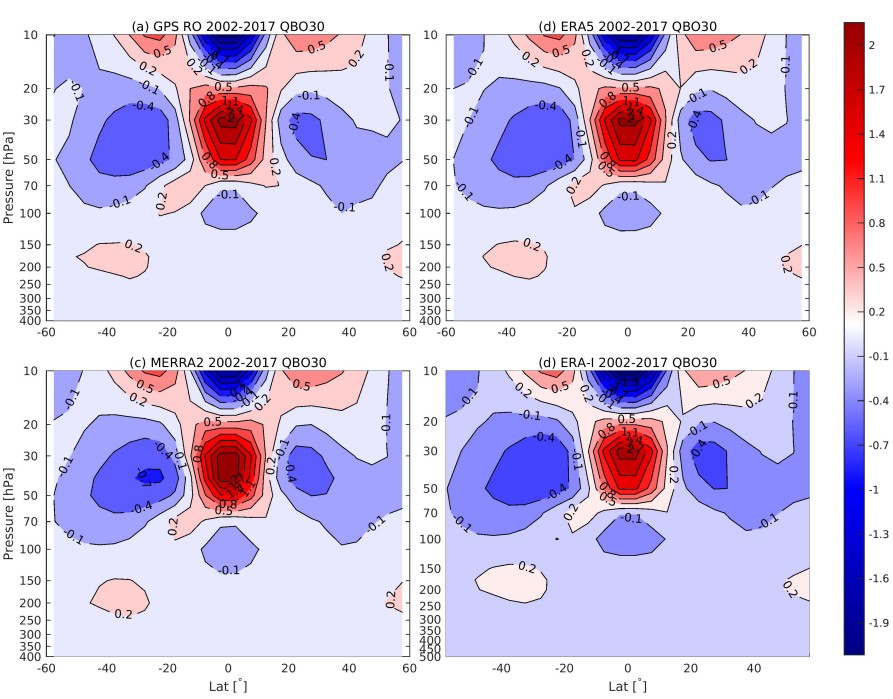

**Figure 10.** Same as figure 8 but for QBO30.





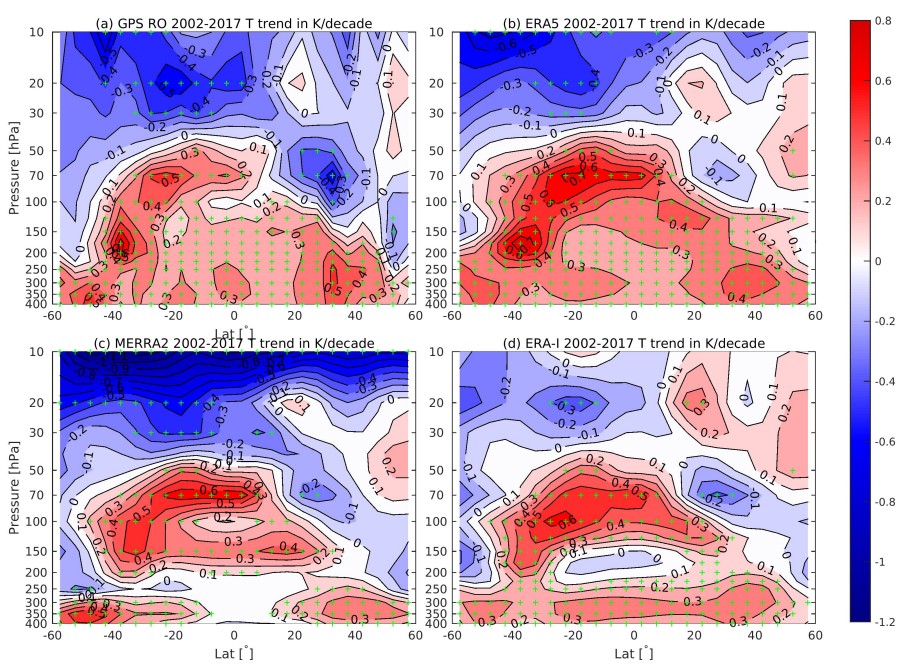

**Figure 11.** Temperature trend in K/decade based on GPS RO (a), ERA5 (b), MERRA2 (c) and ERA-I (d) data for period 2002-2017. The green '+' marked the significant area at 5% level.





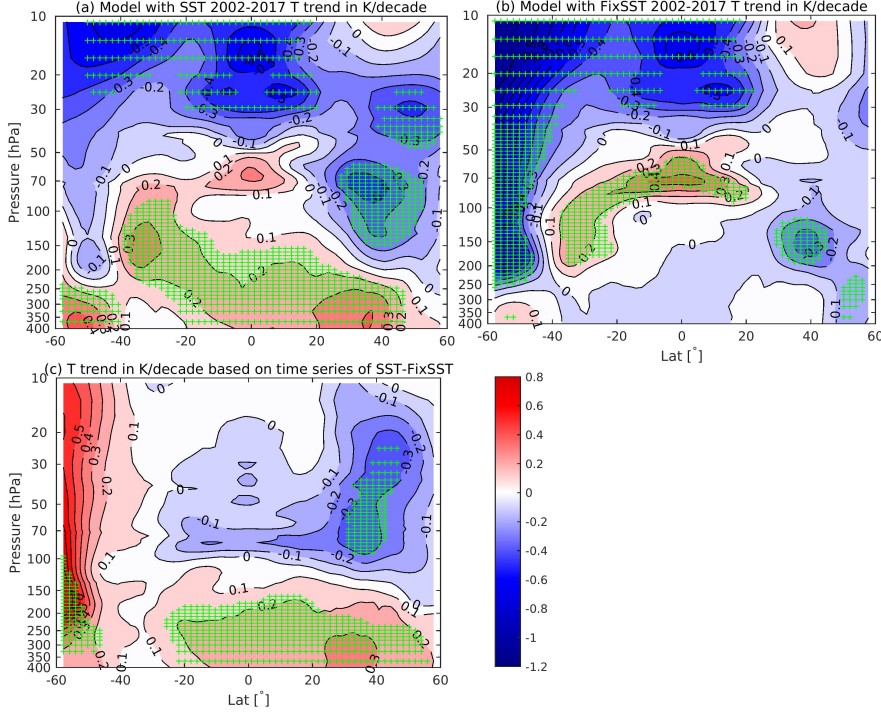

**Figure 12.** Temperature trend in $\mathrm{K/decade}$ based on model simulations with time varying SST (a), fixSST (b) and their differences (c) for period 2002-2017. The green '+' marked the significant area at 5% level.

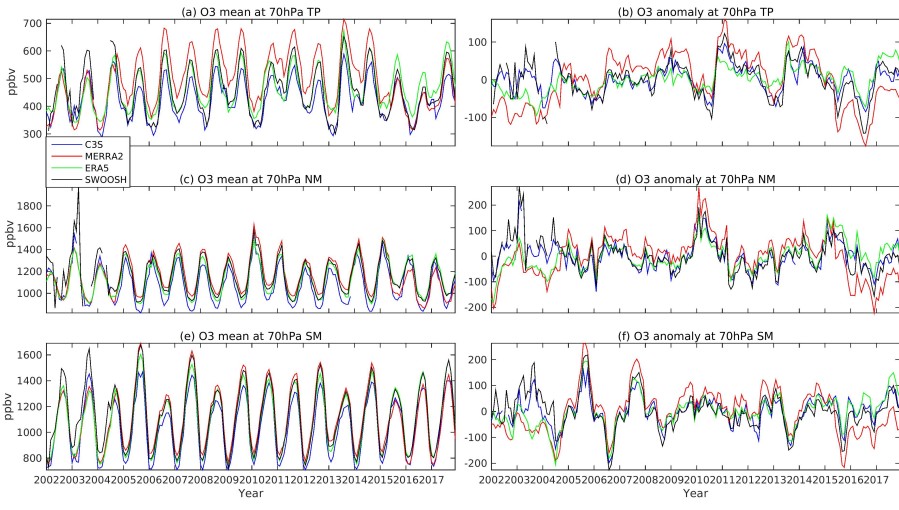

**Figure 13.** Left monthly mean ozone in ppbv at pressure level 70 $\mathrm{hPa}$ through three latitude bands of TP($10°$S-$10°$N)(a), NM($25°$N-$45°$N) (c), NM($25°$S-$45°$S) (e); Right corresponding anomalies in figures (b), (d) and (e); Model with 103 levels (margin), ERA5 (green), C3S (blue), MERRA2 (red) and SWOOSH (black) are included.





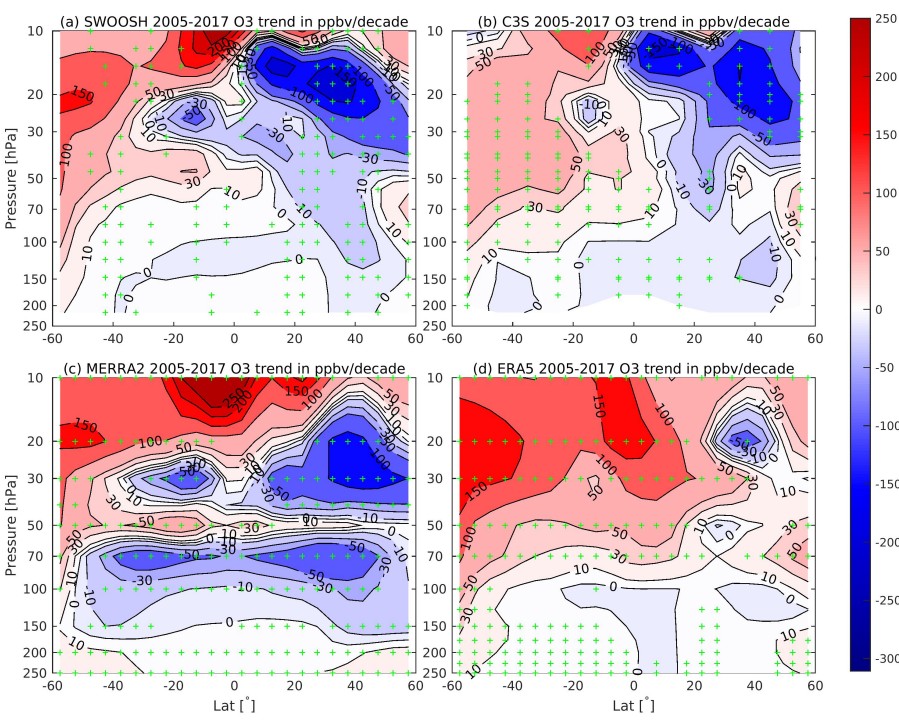

**Figure 14.** Ozone trend in ppbv/decade based on SWOOSH (a), ERA5 (b), MERRA2 (c) and C3S (d) data for period 2005-2017. The green '+' marked the significant area at 5% level.



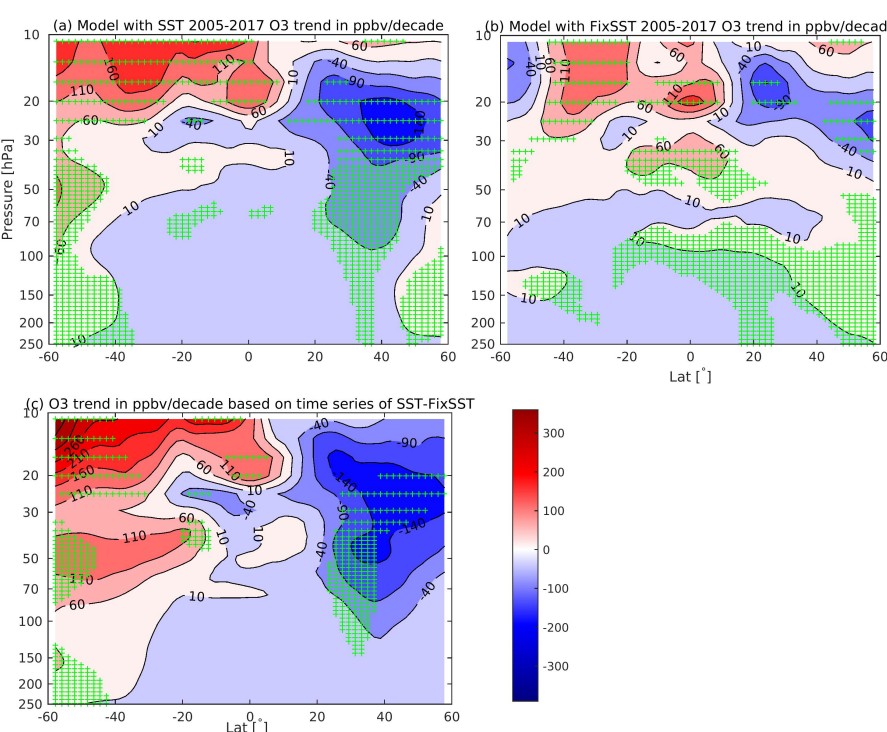

**Figure 15.** Ozone trend in ppbv/decade based on model simulations with time varying SST (a), FixSST (b) and thier differences SST-fixSST (c) for period 2002-2017. The green '+' marked the significant area at 5% level.