# Peer review of "Variability of temperature and ozone in the upper troposphere and lower stratosphere from multi-satellite observations and reanalysis data"

_Atmospheric Chemistry and Physics, 2018_

## Referee Comment (RC1) · Anonymous Referee #1 · 8 Jan 2019

Review of : Variability of temperature and ozone in the upper troposphere and lower stratosphere from multi-satellite observations and reanalysis data. Thank you for the pleasure of reviewing this paper. It is well written (with only a couple of English corrections), well laid out and has very good graphics. I very much like the usage of various satellite data, reanalysis data and comparisons with models to show if/how/where/when reanalysis data could be problematic for trend detection. The usage of a model with and without real time SST shows the role of dynamics and radiation upon the temperature and ozone variations over this albeit short period of 2002-2017. The great thing about GNSS-RO data is that it is unbiased and has been shown by several authors how its assimilation harmonizes the temperatures of the various reanalyses from ∼2004 forward. Unfortunately, as the reanalyses migrate their temperatures toward the GNSS-RO values, any previous bias impacts the temperature trends from before to after the usage of GNSS-RO data. This word of caution is presented by the authors and can't be over emphasized.

I thought the authors do a great job presenting how the ozone and temperatures are interdependent and the roles of dynamics and radiation upon them. The ozone data sets used have both pros and cons. The conversion of number density to ozone mixing ratio is dependent upon the temperatures used. An erroneous trend in temperatures could impart an unwanted trend in ozone mixing ratio values. But that is a separate issue unrelated to the purpose of this paper.

Here are my line-by-line comments:

It is my understanding that GPS-RO is a particular type of radio occultation and that the more general Global Navigation Satellite System (GNSS-RO) should be used to cover all types of RO satellite systems.

Page 3, Line 7: Replace 'get' with 'be'.

Page 3, Line 14: Replace 'continues' with 'continuous'

Page 3, Line 21: Remove the 'a' in the phrase 'found decreasing ozone'

Page 3, Line 25: Is 'LS' defined earlier in the paper, otherwise use 'lower stratosphere'

Page 3, Line 26: Use 'increasing or declining'

Page 3, Line 30: Replace 'Although might be stil problematic' with 'Although it might still be problematic'

Page 4, Line 6: Replace 'recorded' with 'record'

Page 4, Line 12: 'In Sect. 3 we compare'

Page 4, Line 17: Replace 'Around one decade CHAMP' with 'Nearly one decade of

CHAMP'

Page 4, Line 19: Replace 'provides more than 10 times of' with 'providing more than then 10 times the number of'

Page 4, Line 27: Replace 'can be already captured by single satellite' with 'has already been captured by a single satellite'

Page 6, Line 12: 'qulaity' is misspelled 'quality'

Page 6, Line 15: Replace 'since' with 'beginning in'

Page 7, Line 3: Replace 'has been proved for a better representation to the detailed' with 'has been proved to better represent the detailed'

Page 7, Line 19: 'QBO coefficients'

Page 7, Line 20: 'a4' is the QBO30 coefficient, is there a missing solar term in equation 1 with a coefficient a5?

Page 7, Line 30: 'between reanalyses and from the GPS-RO data.'

Page 9, Paragraph beginning at line 3: Does the transition and use of MLS temps affect the MERRA2 trends? How does MERRA2 perform after vs before the use of CHAMP in 2004?

Page 9, Line 31: 'MERRA2', Do you have a reason why ERA-I trends are 'flat'?

Page 10, Line 29: 'estimated'

Page 10, Line 31: 'MERRA2'

Page 11, Line 1: 'At 10 hPa all the data sets'

Page 11, Line 11: 'confirmed by Table 1'

Page 11, Line 28: 'it does not assimilate as many ozone'

[Figure]

Page 12, Line 20: 'which is the reason of the positive'

Page 12, Line 32: 'SST's (Figures 15b-c).'

Page 13, Line 3: 'less ozone in the tropical lower'

Page 13, Line 7: 'SST increases are asymmetric in the two'

Page 13, Line 22: 'shows obvious improvements in reference to ERA-I'

Page 13, Line 23:'well known that are related to'

Page 13, Line 31: 'In contrast to the troposphere'

Page 14, Line 6: 'can be found for the two hemispheres'

Page 14, Line 13: 'supports'

Page 14, Line 17: Remove 'neither'

Figure 1: Label every other year on the X-axis; in the caption: 'between three reanalyses'

Figure 2+: Referring to previous figures should be capitalized: 'Same as Figure 1'

Figure 4: 'The two missions obtained'

Figures 1,2,3,13. It is hard to distinguish the black lines from the blue lines. Could another color or a lighter shade of blue be used?

---

## Referee Comment (RC2) · Anonymous Referee #2 · 2 Feb 2019

Summary and Scientific Contribution:

This paper uses temperature and ozone from satellite measurements and reanalysis products to estimate their variability and trends in the upper troposphere and lower stratosphere (UTLS). Trends are analyzed between 2002 and 2017, and multiple-linear regression model is applied to separate the influences of the Quasi-biennial Oscillation (QBO) and the El Nino Southern Oscillation (ENSO) from trends. In the context of the SPARC Reanalysis Intercomparison Project this paper is an important contribution to the literature. Unfortunately, this paper does not clearly motivate its objective and misses several marks scientifically. In particular, trend analyses over such a short

time-period are suspect and (as the paper shows) inconsistent, making interpretation of these results difficult. Furthermore, connections between ozone and temperature are loosely implied in manuscript without detailed analysis, and the modeling results presented herein are not explained in depth. Finally, the paper is poorly written with grammatical and spelling mistakes throughout, making it very difficult to follow at numerous points. If major revisions are made to address these shortcomings, this paper will be a valuable contribution to the SPARC Reanalysis Intercomparison Project.

Major Comments:

1. This paper is challenging to read because it has significant grammatical errors and spelling mistakes. Often sentences are difficult to parse without several readings, and these problems detract significantly from the scientific content of the paper. For instance, in a part of the paper with an important physically-based discussion (the discussion of model results on pg. 13, line 1), the main sentence of the discussion is so confusing that the message being conveyed is lost. In another example, the primary sentence outlining the paper's goal (pg. 2, line 25) is choppy and unclear, blurring the paper's motivation. I've highlighted some of the more obvious problems in the line-by-line comments below, and at minimum these should be addressed. Preferably, the entire paper would be carefully edited to improve its readability and appropriately convey the authors' scientific findings.

2. Because reanalysis products are combinations of observations and models to assimilate the data, it is disingenuous to consider their trends as directly related to observations. Furthermore, interpretation of reanalysis trends is complicated because the assimilation step brings in data which leads to discontinuities which will vary from place-to-place, time-to-time, and reanalysis-to-reanalysis. The authors themselves acknowledge this problem (pg. 2, line 31), but proceed with their analyses without quantifying how discontinuities affect their results. Reanalyses trend results presented here are suspect and must be interpreted with caution. Without significant changes to the trends analyses (some ideas to do this I suggest below), the authors should instead

shift the main focus of their paper to the comparisons between the variabilities in the reanalysis and GPS products.

3. The problem of interpreting trends from reanalysis is exacerbated by the very short time period considered in this study. A 15-year period (2002-2017) to calculate trends is quite short, and I suspect this contributes to one of the main results of this paper (Table 1), that trends vary in sign and significance depending on the region (except in the tropical middle stratosphere, 10hPa, where trends are more robust, but which is not the focus of this UTLS paper). By eye, the trends appear to be in agreement with one another (Figure 11) in the stratosphere, but there are clear distinctions which makes overall interpretation challenging. This is an inherent difficulty for the study, because GPS data does not extend earlier than 2002. The authors themselves note (citing Santer et al., 2017) that the trend assessment from such short periods can be strongly influenced by start/end years (see also Bandoro et al. 2017, Santer et al. 2011). Given how short the period of record is, without a detailed signal to noise study, is too early to make decisive or defensible claims about UTLS temperature trends in the 21st century. If this study was improved to include a signal-to-noise study which showed the trends are robust, the study results would be more compelling.

4. One of the main reasons short trend calculations here are challenging is because of biases early in the time period (2001-2006), as noted in the text and shown in Figures 1, 2, and 3. These biases early in the period will drive trends in the underlying data which will factor into the trends calculated with the MLR method. For instance, I can quickly estimate the following trends in the biases: @400hPa: +0.2 K/decade, @100hPa: +0.35 K/decade, @70hPa: +0.25 K/decade. Each of these is on the order of the trends found in Table 1 for those regions, making it very difficult to determine whether trends found to be "significant" are actually just trending because of early period biases. Table 1 should be updated to include the trends in the biases (like the estimates above) for each product and region (or some similar analysis), and to directly with the calculated trends (e.g., this method is used to examine radiosonde trends in Wang et al. 2012).

Where the bias trend is on the order of the product temperature trends, the robustness of those trends should be reconsidered.

5. The residuals and the anomalies of the multivariate regression (Figures 6 and 7) have same exact temporal structure and nearly the same magnitude. Do you know why? Can you directly compare and contrast your results with those of Randel and Wu (2014) who completed a detailed analysis using this method? It is concerning that the residuals have a magnitude that is roughly the same as the signal, suggesting the majority of the signal is unexplained (e.g. QBO and ENSO both have amplitudes of less than 0.05K at this height).

6. Another concern I have with this study is that the connections between ozone and temperature are very loosely made, and there are no analyses to support them. Calculations (such as changes in temperature structure through changes in ozone through either a climate model or radiative transfer model) have not been made, and not even a simple correlation analysis was performed. Many previous studies (e.g. Abalos et al. 2012, Maycock 2016, Gilford et al. 2016, to name just a few) have done detailed modeling, radiative calculations or statistical analyses, quantifying the relationship between temperature and ozone. Instead, this paper simply notes "In the stratosphere, ozone distribution is highly correlated with the temperature change" (pg. 14, line 3) without actually showing any such correlations, and discusses some loose connections between temperature and ozone in section 3.4. Furthermore, it claims we need to "await further investigation" (pg. 3, line 27), but extensive research on this topic has been done! There is very little acknowledgement of the vast literature which has discussed this topic in detail, and the results herein are not framed within that context. Its important to perform some analysis to show how this work is valuable and contributing to our knowledge of ozone/temperature links (especially in the context of how this relationship changes between reanalyses and GPS).

7. My primary concern with this paper is that it does not successfully and clearly distinguishing itself as novel. The trend calculations (for instance for ozone, pg. 3, line

21) have been updated through 2016 in previous studies, so this paper represents a 2-year improvement (and as noted above, the depth ozone research herein is not at a level commensurate with previous studies). Studies of UTLS temperature variability from GPS measurements have been very robustly presented in previous works (e.g. Abalos et al. 2012, Randel and Wu 2014). The use of the model to explore these processes is not well explained in the text, or compared with recently published studies which have done this (e.g. Randel et al. 2017).

To address this, I recommend the authors realign their motivation, highlighting that they are primarily concerned with comparing reanalyses and GPS in the UTLS with ERA5, in accordance with the S-RIP. Improvements in the ozone analyses and trend bias estimations in the context of comparing reanalyses will further improve on this narrative. Furthermore, the model should be brought introduced earlier in the paper as part of the motivation. This study can and will be valuable, but you need the tell and show the readers in clear language!

Figure Comments:

All Figures: Please include units in all of your figure captions and titles/axes (where relevant).

Figure 1: One of the ranges in the caption should be "SM" instead of "NM". Also, it is not explained anywhere what is meant by SM and NM. Please add an explanation in the text of the manuscript.

Figures 4-5, 8-12, 14-15: Zonal mean figures would be improved if a line was added to indicate the climatological zonal mean tropopause height (using either the lapse rate tropopause or the cold-point tropopause, see Munchak and Pan 2014). These will likely vary from product to product and in the model, but it will help the read understand how your results vary with respect to the tropopause height.

Figures 1-3, 13: The x-axes on these timeseries plots are very hard to read because

the years are all squished together.

Figures 11-12, 14-15 (and timeseries plots): Readers who are green-red will find it very difficult to parse the green "+" markers or green lines in these figures. Please use some other way or color contrast this data which is color-blind friendly.

Table 1: This is a key result in the entire paper, yet its unclear. What are the +/- values in this table, are they the confidence intervals from your t-test? If so, please indicate so. It's also important that trends in the biases from GPS RO be included as a column at each level, for comparison.

Line-By-Line Comments:

Pg. 1, line 1: This first sentence is confusing as written.

Pg. 1, line 2 and elsewhere: Replace "were" with "are", and use present tense language throughout.

Pg. 1, line 3+15: The first few sentences need to motivate the reader as to why your study is a valuable contribution and novel. I recommend mentioning the model here in addition to later, and be specific about what model you are using and in what mode.

Pg. 1, line 13: replace "the change of" with "discontinuities in"

Pg. 1, line 16: The use of "could be" shows how the shallow the ozone and physically-based analyses in this study are. Further analyses should allow you to be more definitive here.

Pg. 2, line 1: It is not "the" key region, it is "a" key region. Coupling is also important at high latitudes (e.g. sudden stratospheric warmings).

Pg. 2, line 3: Do you mean that temperatures in the UTLS respond to climate change? That they affect other things (like water vapor) so they indirectly affect climate change? Please rewrite for clarity.

Pg. 2, lines 7-9: This sentence is confusing and should be rewritten.

Pg. 2, line 9: "through" should be "between"

Pg. 2, line 11: The term "underlying mechanisms" is used 4 times in this text without any clear explanation of what it means. Its use is vague and unspecific, please rewrite to clarify exactly what is meant when you say "underlying mechanisms".

Pg. 2, line 11: You are talking about trends in this paragraph, but now you mention variability (which could be construed as interannual variability). important to keep them distinct throughout the paper, because they could be changing in different ways.

Pg. 2, line 24: This is very poorly written sentence, please rewrite for clarity.

Pg. 2, line 27: "Plenty" is a slang term and not professional. Please look throughout your manuscript and replace these slang terms with more specific ones (e.g. "On one hand", pg. 3, line 4; "Same as", pg. 6, line 24; etc.). Here I suggest: "assimilate ground-based, satellite-based, and other data sources to provide the current..."

Pg. 2, line 31: The use of "perform" here is not correct. "may exhibit" would work. Other times in this paper "perform" is also not used correctly (e.g. pg. 13, line 24); please rewrite each of these.

Pg. 3, lines 1-2: This sentence is poorly written and distorts the communication of your goal.

Pg. 3, line 9: While ozone changes could be a helpful indicator as you claim, you've barely touched on how complicated this is. Schoeberl et al. (2008) did a rather complete study of this, but others (e.g. Polvani and Solomon 2012) have shown that it has rich nuances. You skip over that richness in your literature review here. I think its worth noting the efforts those papers made, and how your work is different.

Pg. 3, line 10: "various of" should be "various"

Pg. 3, line 17: Very confusing as written.

Pg. 3, line 19: 15 hPa is well above the UTLS region!

Pg. 3, line 29: The sentence is confusing as written.

Pg. 3, line 34: This a very abrupt transition introducing the model. This needs to be done more smoothly and with better motivation as to why we are using the model.

Pg. 4, lines 3-10: Much of this paragraph is repetive with previous ones and can be removed.

Pg. 4, line 10: What is meant by "dynamical processing with SST"?

Pg. 4, line 17: Seven years is not one decade. This is also very confusing as written.

Pg. 4, line 22: Are these measurement errors? Or differences from some other instrument?

Pg. 4, line 34: Can you provide a magnitude estimate for this "low effect"?

Pg. 5, line 14: Was this linear interpolation done on a pressure grid or a height grid?

Pg. 5, line 17: What is meant by comparable here?

Pg. 5, line 25: add "to" before "which"

Pg. line 27: There's no transition between these paragraphs. Are you introducing a new dataset you will also use?

Pg. 6, line 2: On what basis can you call this "a time period suitable for trend evaluation"?

Pg. 6, line 7: introduce this as version 3 in the very first sentence of this paragraph instead.

Pg. 6, line 16: As written, this sentence is unreadable. I don't understand what it is trying to say.

Pg. 6, line 20: The link doesn't work as written, and should be more carefully cited in

the bibliography.

Pg. 7, line 10: Please rewrite this confusing sentence.

Pg. 7, line 11: I recommend renaming this section "Trend Calculations"

Pg. 7, line 15: "Phenomenons" should be "phenomena"

Pg. 7, line 20: You have "a4" twice, but no solar component in equation 1.

Pg. 7, line 25: Is this a one-sided or two-sided t-test? Also, is this significance level the p-value? Please clarify your method.

Pg. 7, line 29: The 400hPa level is well below the tropopause, especially in the tropics.

Pg. 8 line 11: What do you mean by "more disturbed" here?

Pg. 9, line 22: why does the shortness of the period change this result? The shorter period means that interannual variability should have more influence on the trend calculations.

Pg. 9, line 27: "getting less" should be "smaller"

Pg. 9, line 29: The sentence is very confusing as written.

Pg. 10, lines 4 and 12: What phase of ENSO or QBO? Please clarify throughout your paper what phase you mean each time you discuss results for QBO and ENSO.

Pg. 10, line 17: This title isn't worded correctly. I suggest "Temperature Trends"

Pg. 10, line 28: I don't know what you mean by this sentence, you might be missing a word?

Pg. 10, line 31: "MEERA2" should be "MERRA2".

Pg. 11, line 5: Which tropopause? The cold point? The tropopause is a transition layer in the tropics (Fueglistaler et al. 2009).

Pg. 11, line 17: what dynamic process do you mean? Do you mean the influences of SSTs on circulation? If so, please say so.

Pg. 11, line 28: "so many" should be "as many"

Pg. 12, line 35: This is a nice physical discussion which is mired by very unclear writing.

Pg. 13, line 1: Can you cite this? Many papers have shown this result.

Pg. 13, line 3: "That is not the truth" is not professional; please rewrite.

Pg. 13, line 5: There is no observational evidence for ozone recovery yet, outside the spring SH stratosphere (Randel et al. 2017).

Pg. 13, line 16: You haven't done any attribution work, so this claim should be removed.

Pg. 13, line 22-24: These lines are very confusing; I don't understand what you mean.

Pg. 13, line 29: 15 years is not "nearly 2 decades".

Pg. 14, line 1: This is a run-on sentence, and its very hard to parse what your point is here. Please rewrite.

Pg. 14, lines 3: You have not shown this result.

Pg. 14, line 5: This result isn't true for all datasets in your study, and you haven't clarified what period these trends are considered over in this discussion.

Pg. 14, line 14: Your results do not show this link, please don't make false claims without evidence. In fact, it has been shown previously to not be the case (Randel et al. 2017).

Pg. 14, line 17: Poorly written.

References:

Abalos et al. (2012): Variability in upwelling across the tropical tropopause and corre-

lations with tracers in the lower stratosphere. Atmos. Chem. Phys., 12, 11505–11517, www.atmos-chem-phys.net/12/11505/2012/ doi:10.5194/acp-12-11505-2012

Bandoro et al. (2018): Detectability of the impacts of ozone-depleting substances and greenhouse gases upon stratospheric ozone accounting for nonlinearities in historical forcings. Atmos. Chem. Phys., 18, 143–166, https://doi.org/10.5194/acp-18-143-2018.

Fueglistaler, S., A. E. Dessler, T. J. Dunkerton, I. Folkins, Q. Fu, and P. W. Mote, 2009: Tropical tropopause layer. Rev. Geophys., 47, RG1004, doi:10.1029/2008RG000267.

Gilford, D. M., S. Solomon, and R. W. Portmann, 2016: Radiative impacts of the 2011 abrupt drops in water vapor and ozone in the tropical tropopause layer. J. Climate, 29, 595–612, doi:10.1175/JCLI-D-15-0167.1.

Jiang, J. H., H. Su, C. Zhai, L. Wu, K. Minschwaner, A. M. Molod, and A. M. Tompkins (2015), An assessment of upper troposphere and lower stratosphere water vapor in MERRA, MERRA2, and ECMWF reanalyses using Aura MLS observations, J. Geophys. Res. Atmos., 120, 11,468–11,485, doi:10.1002/ 2015JD023752

Maycock, A. C. (2016), The contribution of ozone to future stratospheric temperature trends, Geophys. Res. Lett., 43, doi:10.1002/2016GL068511.

Munchak, L. A., and L. L. Pan (2014), Separation of the lapse rate and the cold point tropopauses in the tropics and the resulting impact on cloud top-tropopause relationships, J. Geophys. Res. Atmos., 119, 7963–7978, doi:10.1002/ 2013JD021189.

Polvani, L. M., and S. Solomon (2012), The signature ofozone depletion on tropical temperature trends, as revealed by their seasonal cycle in model integrations with single forcings, J. Geophys. Res., 117, D17102, doi:10.1029/2012JD017719.

Randel and Wu (2014): Variability of zonal mean tropical temperatures derived from a decade of GPS radio occultation data. JAS, doi: 10.1175/JAS-D-14-0216.1.

Randel, W. J., Polvani, L., Wu, F., Kinnison, D. E., Zou, C.-Z., & Mears, C. (2017).

[Figure]

Troposphere-stratosphere temperature trends derived from satellite data compared with ensemble simulations from WACCM. Journal of Geophysical Research: Atmospheres, 122. https://doi. org/10.1002/2017JD027158

Santer, B. D., et al. (2011), Separating signal and noise in atmospheric temperature changes: The importance of timescale, J. Geophys. Res. ,116, D22105, doi:10.1029/2011JD016263.

Schoeberl, M. R., et al. (2008), QBO and annual cycle variations in tropical lower stratosphere trace gases from HALOE and Aura MLS observations, J. Geophys. Res., 113,D05301, doi:10.1029/2007JD008678.

Wang, J. S., D. J. Seidel, and M. Free (2012), How well do we know recent climate trends at the tropical tropopause?, J. Geophys. Res., 117, D09118, doi:10.1029/2012JD017444.

---

## Short Comment (SC1) · 9 Feb 2019

Dear Authors

Please consider these two comments, one scientific and one regarding data citation. This is an interesting paper and I hope you will find my remarks helpful.

1) I really appreciate your discussion of the negative impacts of step-changes in the ozone observing system on ozone trends in MERRA-2. Even a cursory look at Figure 13 reveals that the discontinuity associated with the transition from MLS v2.2 to v4.2 in June 2015 is nontrivial, as you correctly point out in Section 3.4. I would like to draw

your attention to the fact that it is possible and relatively simple to account for this, as well as the 2004 SBUV-to-MLS transition, precisely because these step-changes are so infrequent and well defined. In Wargan et al, 2018 (doi:10.1029/2018GL077406) we did it using an SD model simulation as a transfer function but it could also be done by including a step-function proxy in the MLR. We tried the latter approach (not shown in our paper) and the result was very similar to that obtained using the transfer function approach. I suspect the MERRA-2 panel in Figure 14 would look different if a bias correction was applied. In fact, the analysis could be extended further back to 1998.

2) NASA GMAO asks the users of MERRA-2 data to explicitly cite the data collections used. Note that each MERRA-2 collection has a unique doi number listed in the file specs document https://gmao.gsfc.nasa.gov/pubs/docs/Bosilovich785.pdf

For example monthly mean pressure-levels assimilated data ("M2IMNPASM" or *in-stM_3d_asm_Np*) could be cited as follows:

Global Modeling and Assimilation Office (GMAO) (2015), MERRA-2 instM_3d_asm_Np: 3d,Monthly mean,Instantaneous,Pressure-Level,Assimilation,Assimilated Meteorological Fields V5.12.4, Greenbelt, MD, USA, Goddard Earth Sciences Data and Information Services Center (GES DISC), Accessed: [Data Access Date], 10.5067/2E096JV59PK7

I hope this is helpful and thanks for using MERRA-2! Best regards, Kris Wargan

Reference Wargan, K., Orbe, C., Pawson, S., Ziemke, J. R., Oman, L. D., Olsen, M. A., et al. (2018). Recent decline in extratropical lower stratospheric ozone attributed to circulation changes. Geophysical Research Letters, 45, 5166–5176. https://doi.org/10.1029/2018GL077406
* * *

---

## Author Comment (AC1) · 27 Mar 2019

**Reviewer #1 comment on "Variability of temperature and ozone in the upper troposphere and lower stratosphere from multi-satellite observations and reanalysis data" by Shangguan et. al.**

Reviewer #1 (Comments to Author):
Review of: Variability of temperature and ozone in the upper troposphere and lower stratosphere from multi-satellite observations and reanalysis data. Thank you for the pleasure of reviewing this paper. It is well written (with only a couple of English corrections), well laid out and has very good graphics. I very much like the usage of various satellite data, reanalysis data and comparisons with models to show if how/where/when reanalysis data could be problematic for trend detection. The usage of a model with and without real time SST shows the role of dynamics and radiation upon the temperature and ozone variations over this albeit short period of 2002-2017. The great thing about GNSS-RO data is that it is unbiased and has been shown by several authors how its assimilation harmonizes the temperatures of the various reanalyses from ~2004 forward. Unfortunately, as the reanalyses migrate their temperatures toward the GNSS-RO values, any previous bias impacts the temperature trends from before to after the usage of GNSS-RO data. This word of caution is presented by the authors and can't be over emphasized.
I thought the authors do a great job presenting how the ozone and temperatures are interdependent and the roles of dynamics and radiation upon them. The ozone data sets used have both pros and cons. The conversion of number density to ozone mixing ratio is dependent upon the temperatures used. An erroneous trend in temperatures could impart an unwanted trend in ozone mixing ratio values. But that is a separate issue unrelated to the purpose of this paper.
Thank you very much for your very helpful comments. We have revised our manuscript accordingly and hope the manuscript have been considerably improved. Please see our point-to-point response as follows.
Reviewer comments are in black, following by our respective replies in blue.

King regards,

Ming Shangguan (on behalf of all co-authors)
Here are my line-by-line comments:

It is my understanding that GPS-RO is a particular type of radio occultation and that the more general Global Navigation Satellite System (GNSS-RO) should be used to cover all types of RO satellite systems.
Thank you very much for your remark. We have modified all the GPS-RO to GNSS-RO in the text.
Page 3, Line 7: Replace 'get' with 'be'.
It has been corrected.
Page 3, Line 14: Replace 'continues' with 'continuous'
Corrected.

Page 3, Line 21: Remove the 'a' in the phrase 'found decreasing ozone'

Updated.

Page 3, Line 25: Is 'LS' defined earlier in the paper, otherwise use 'lower stratosphere'

Thank you for your remark. We have replaced LS with the lower stratosphere.

Page 3, Line 26: Use 'increasing or declining'

Done.

Page 3, Line 30: Replace 'Although might be still problematic' with 'Although it might still be problematic'

Corrected.

Page 4, Line 6: Replace 'recorded' with 'record'

Done.

Page 4, Line 12: 'In Sect. 3 we compare'

Updated.

Page 4, Line 17: Replace 'Around one decade CHAMP' with 'Nearly one decade of CHAMP'

Thanks. We have updated this sentence as suggested.

Page 4, Line 19: Replace 'provides more than 10 times of' with 'providing more than 10 times the number of'

Yes, it has been done.

Page 4, Line 27: Replace 'can be already captured by single satellite' with 'has already been captured by a single satellite'

Thanks. We have revised this sentence as suggested.

Page 6, Line 12: 'qulaity' is misspelled 'quality'

Corrected.

Page 6, Line 15: Replace 'since' with 'beginning in'

We have done the correction according to your comment.

Page 7, Line 3: Replace 'has been proved for a better representation to the detailed' with 'has been proved to better represent the detailed'.

Done.

Page 7, Line 19: 'QBO coefficients'

Corrected.

Page 7, Line 20: 'a4' is the QBO30 coefficient, is there a missing solar term in equation1 with a coefficient a5?

We apologize for this mistake. Yes, 'a4' is the QBO30 coefficient. We have corrected this sentence in the revised manuscript.

Page 7, Line 30: 'between reanalyses and from the GPS-RO data.'

Corrected.

Page 9, Paragraph beginning at line 3: Does the transition and use of MLS temps affect the MERRA2 trends? How does MERRA2 perform after vs before the use of CHAMP in 2004?

According to previous studies (e.g., McCarty et al., 2016; Fujiwara et al., 2017), the MERRA2 only assimilated MLS temperature observations at and above 5 hPa. For this study, since we are focusing on the region below 10 hPa, the MERRA2 trends

shown in this study should not be affected by the MLS temperatures. However, this effect should be considered while investigating temperature trends above 5 hPa. The effect of the CHAMP to MERRA2, as introduced in MERRA2 on 15 July 2004 (McCarty et al., 2016), is not significant since the single CHAMP satellite has very limited number of observations.

Page 9, Line 31: 'MERRA2', Do you have a reason why ERA-I trends are 'flat'?

As shown in Figure 6(a) the ERA-I temperature anomalies from 2002 to mid-2006 are highest compared to other data sets. According to Simmons et al. (2014), local degradation occurs near the sub-tropical tropopause whereas substantial amounts of warm-biased aircraft data are assimilated since 1999. After 2006, while large number of COSMIC data is assimilated, this warm bias disappeared. This led to less warming at 150 hPa in the tropical region represented by ERA-I.

Page 10, Line 29: 'estimated'

Corrected.

Page 10, Line 31: 'MERRA2'

Corrected.

Page 11, Line 1: 'At 10 hPa all the data sets'

Updated.

Page 11, Line 11: 'confirmed by Table 1'

Done.

Page 11, Line 28: 'it does not assimilate as many ozone'

Corrected.

Page 12, Line 20: 'which is the reason of the positive'

Done.

Page 12, Line 32: 'SST's (Figures 15b-c).'

Corrected.

Page 13, Line 3: 'less ozone in the tropical lower'

Done.

Page 13, Line 7: 'SST increases are asymmetric in the two'

We have done this update.

Page 13, Line 22: 'shows obvious improvements in reference to ERA-I'

Corrected.

Page 13, Line 23:'well known that are related to'

Done.

Page 13, Line 31: 'In contrast to the troposphere'

Done.

Page 14, Line 6: 'can be found for the two hemispheres'

Done.

Page 14, Line 13: 'supports'

Corrected.

Page 14, Line 17: Remove 'neither'

Done.

Figure 1: Label every other year on the X-axis; in the caption: 'between three reanalyses'

Thanks. We have updated the figure as well as the caption as suggested.

Figure 2+: Referring to previous figures should be capitalized: 'Same as Figure 1'

Thanks, it has been corrected.

Figure 4: 'The two missions obtained'

Done.

Figures 1,2,3,13. It is hard to distinguish the black lines from the blue lines. Could another color or a lighter shade of blue be use.

Thank you very much for your advices. We have changed the color in Figures 1, 2, 3,13.

References

McCarty, W., Coy, L., Gelaro, R., Huang, A., Merkova, D., Smith, E. B., Sienkiewicz, M., and K., W.: NASA Tech. Rep. NASA/TM{2016{104606, https://gmao.gsfc.nasa.gov/pubs/docs/McCarty885.pdf, 2016.

Fujiwara, M., Wright, J. S., Manney, G. L., Gray, L. J., Anstey, J., Birner, T., Davis, S., Gerber, E. P., Harvey, V. L., Hegglin, M. I., Homeyer, C. R., Knox, J. A., Kruger, K., Lambert, A., Long, C. S., Martineau, P., Molod, A., Monge-Sanz, B. ¨M., San- tee, M. L., Tegtmeier, S., Chabrillat, S., Tan, D. G. H., Jack- son, D. R., Polavarapu, S., Compo, G. P., Dragani, R., Ebisuzaki, W., Harada, Y., Kobayashi, C., McCarty, W., Onogi, K., Paw- son, S., Simmons, A., Wargan, K., Whitaker, J. S., and Zou, C.-Z.: Introduction to the SPARC Reanalysis Intercomparison Project (S-RIP) and overview of the reanalysis systems, At- mos. Chem. Phys., 17, 1417{1452, https://doi.org/10.5194/acp- 17-1417-2017, 2017.

Simmons, A. J., Poli, P., Dee, D. P., Berrisford, P., Hersbach, H., Kobayashi, S., and Peubey, C.: Estimating low-frequency variability and trends in atmospheric temperature using ERA-Interim, Quarterly Journal of the Royal Meteorological Society, 140,329-353, 2014.

---

## Author Comment (AC2) · 27 Mar 2019

**Reviewer #2 comment on "Variability of temperature and ozone in the upper troposphere and lower stratosphere from multi-satellite observations and reanalysis data" by Shangguan et. al.**

Reviewer #2 (Comments to Author):

This paper uses temperature and ozone from satellite measurements and reanalysis products to estimate their variability and trends in the upper troposphere and lower stratosphere (UTLS). Trends are analyzed between 2002 and 2017, and multiple-linear regression model is applied to separate the influences of the Quasi-biennial Oscillation (QBO) and the El Nino Southern Oscillation (ENSO) from trends. In the context of the SPARC Reanalysis Intercomparison Project this paper is an important contribution to the literature. Unfortunately, this paper does not clearly motivate its objective and misses several marks scientifically. In particular, trend analyses over such a short time-period are suspect and (as the paper shows) inconsistent, making interpretation of these results difficult. Furthermore, connections between ozone and temperature are loosely implied in manuscript without detailed analysis, and the modeling results presented herein are not explained in depth. Finally, the paper is poorly written with grammatical and spelling mistakes throughout, making it very difficult to follow at numerous points. If major revisions are made to address these shortcomings, this paper will be a valuable contribution to the SPARC Reanalysis Intercomparison Project.

We thank the reviewer very much for the very constructive and useful comments and suggestions. We have revised the manuscript according to all the comments. Firstly, we have rewritten our introduction to explain our motivation clearly in the context of the SPARC Reanalysis Intercomparison Project. Secondly, we rechecked the significance of the trends by calculating the signal-to-noise ratio. Thirdly, we have made a correlation test between temperature and ozone time series to study the connection between ozone and temperature. We apologize for the grammatical and spelling mistakes and we have checked the whole text carefully and corrected the mistakes. We hope the reviewer could find the manuscript has been improved significantly.

Please see below our point-to-point response to all reviewers' comments and suggestions. Reviewer comments are in black, following by our respective replies in blue.

Kind regards,

Ming Shangguan (on behalf of all co-authors)

Major Comments:

1. This paper is challenging to read because it has significant grammatical errors and spelling mistakes. Often sentences are difficult to parse without several readings, and these problems detract significantly from the scientific content of the paper. For instance, in a part of the paper with an important physically-based discussion (the discussion of model results on pg. 13, line 1), the main sentence of the discussion is so confusing that the message being conveyed is lost. In another example, the primary sentence outlining the paper's goal (pg. 2, line 25) is choppy and unclear, blurring the paper's motivation. I've highlighted some of the more obvious problems in the line-by-line comments below, and at minimum these should be addressed. Preferably, the entire paper would be carefully edited to improve its readability and appropriately convey the authors' scientific findings.

Thank you very much for your comments. We are really sorry for so many grammatical errors and spelling mistakes in the text. We have modified the text according to your suggestions and edited the entire paper carefully. The introduction has been rewritten to explain our motivation clearly. More details can be found in our line-by-line response and the revised manuscript.

2. Because reanalysis products are combinations of observations and models to assimilate the data, it is disingenuous to consider their trends as directly related to observations. Furthermore, interpretation of reanalysis trends is complicated because the assimilation step brings in data which leads to discontinuities which will vary from place-to-place, time-to-time, and reanalysis-to-reanalysis. The authors themselves acknowledge this problem (pg. 2, line 31), but proceed with their analyses without quantifying how discontinuities affect their results. Reanalyses trend results presented here are suspect and must be interpreted with caution. Without significant changes to the trends analyses (some ideas to do this I suggest below), the authors should instead shift the main focus of their paper to the comparisons between the variabilities in the reanalysis and GPS products.

We totally agree with the reviewer that the reanalysis products are influenced by both observations and assimilation systems and should not be compared to observed trends directly. According to your suggestions, we have rewritten our introduction and shift the main focus of the paper to the comparisons between the variabilities in the reanalysis and GNSS products. In addition, we corrected the temperature discontinuities around 2006 in the reanalysis by using a transfer function approach similar to Wargan et al., 2018. The corrected GNSS RO time series was used as a common baseline since it does not have significant discontinuities. Details of the bias correction for reanalysis temperatures can be seen in the supplementary information. The temperature trends from reanalysis data sets were recalculated and their significance was also rechecked using the signal-to-noise ratio.

3. The problem of interpreting trends from reanalysis is exacerbated by the very short time period considered in this study. A 15-year period (2002-2017) to calculate trends is quite short, and I suspect this contributes to one of the main results of this paper (Table 1), that trends vary in sign and significance depending on the region (except in the tropical middle stratosphere, 10hPa, where trends are more robust, but which is not the focus of this UTLS paper). By eye, the trends appear to be in agreement with one another (Figure 11) in the stratosphere, but there are clear distinctions which makes overall interpretation challenging. This is an inherent difficulty for the study, because GPS data does not extend earlier than 2002. The authors themselves note (citing Santer et al., 2017) that the trend assessment from such short periods can be strongly influenced by start/end years (see also Bandoro et al. 2017, Santer et al. 2011). Given how short the period of record is, without a detailed signal to noise study, is too early to make decisive or defensible claims about UTLS temperature trends in the 21st century. If this study was improved to include a signal-to-noise study which showed the trends are robust, the study results would be more compelling.

Thank you very much for the constructive comments. Yes, the 16-year period is relatively short to calculate trends and there is clear distinction between different data sets especially in regions with insignificant trends. According to your suggestion we have made a signal to noise study based on three 145-years CESM simulations. The CESM runs were integrated in a fully coupled mode with an interactive ocean for the time period 1955 to 2099. All anthropogenic forcing, e.g. GHGs and ODSs were fixed to values at the year 1960. The three simulations are slightly different with the natural forcing. The first run used observed solar irradiance, time varying volcanic aerosols and a nudged QBO, while the second run fixed the solar irradiance as a constant and the third run did not include a QBO. More details of the simulations can be seen in the supplementary information. The influences of solar cycle, volcanic aerosols and QBO were excluded by a multiple linear regression before the calculations of the background noise.

To assess the effect of seasonal and interannual variability on 16-year temperature trends, we fit linear trends to overlapping 192-month segments of the 1740-month in each of CESM runs. For maximally overlapping 192-month intervals (i.e., for overlap by all but one month), one simulation yields 1549 samples of 192-month trends. Following the method described by Bandoro et al. 2017 and Santer et al. 2011, we exclude the largest cooling or warming trends from our analysis and calculate the standard deviations of the 16-year trends (right panel in Fig.1). Note that the method used here is slightly different with that in Bandoro et al. 2017. We estimated the standard deviation of by different overlapping 16-year trends from the same model while they used a large ensemble of simulations with different models. The advantage of their approach is that the results are not model dependent. However, our results based on the CESM model should be helpful since it is one of the best models and has been widely used in UTLS studies.

The signal to noise ratios of 16-year GNSS RO temperature trends are shown in Fig.1 (left panel). Here we use the 90% and 95% significance level, which corresponds to a signal to noise ratio close to 1.65 and 1.96. Seen from Fig. 1, the areas with significant trends are smaller than that shown in Fig. 11 in the main text. However, there are still significant signals in the mid-latitudes of the upper troposphere, around the tropopause and in the southern hemisphere in the middle stratosphere. All the significant regions in Fig. 1 are actually the most important areas with strongest and significant trends in Fig. 11. This suggests that the significant trends shown in Fig. 11 are robust except that in the tropics whereas the standard deviation of the trends are the strongest.

To my understanding, the signal-to-noise ratio suggested by the reviewer and the significance test used in this manuscript are actually two methods to test the significance/robustness of the calculated trends. The main difference between the two methods is the way to estimate the standard deviation/noise. Since the standard deviation of the residuals of the linear fit has been widely used in trend analysis (e.g., Wigley et al., 2006), we would like to keep the significance test as it was in the manuscript. At the same time, we have put Fig. 1 in the supplementary and added some discussions correspondingly in the revised manuscript.

[Figure]

Figure 1: Signal to noise ratios (left) are estimated RO trends divided by the standard deviations of model trends (right), calculated using overlapping time series segment.

4. One of the main reasons short trend calculations here are challenging is because of biases early in the time period (2001-2006), as noted in the text and shown in Figures 1. These biases early in the period will drive trends in the underlying data which will factor into the trends calculated with the MLR method. For instance, I can quickly estimate the following trends in the biases: @400hPa: +0.2 K/decade, @100hPa: +0.35 K/decade, @70hPa: +0.25 K/decade. Each of these is on the order of the trends found in Table 1 for those regions, making it very difficult to determine whether trends found to be "significant" are actually just trending because of early period biases. Table1 should be updated to include the trends in the biases (like the estimates above) for each product and region (or some similar analysis), and to directly with the calculated trends (e.g., this method is used to examine radiosonde

trends in Wang et al. 2012). Where the bias trend is on the order of the product temperature trends, the robustness of those trends should be reconsidered.

Thank you for your suggestion. We tried to add the bias trends in table 1. However, there are too many numbers and hard to clearly show the important information. Therefore, we put the uncorrected and corrected trends in a Figure similar to Wang et al. 2012. We use the following figure instead of Table 1 in the revised manuscript. The impacts of biases on calculated trends are also discussed in the text.

[Figure]

Figure 2: Estimated temperature trends in K/decade in different regions (SM: 25 ˚S-45 ˚S; NM: 25 ˚N-45 ˚N; TP: 10 ˚S-10 ˚N) from 2002 to 2017. (a-f) Trends in corrected and uncorrected data sets at 250, 150, 70, 50, 20 and 10 hPa. Error bars represent 95% confidence intervals.

5. The residuals and the anomalies of the multivariate regression (Figures 6 and 7) have same exact temporal structure and nearly the same magnitude. Do you know why? Can you directly compare and contrast your results with those of Randel and Wu (2014) who completed a detailed analysis using this method? It is concerning that the residuals have a magnitude that is roughly the same as the signal, suggesting the majority of the signal is unexplained (e.g. QBO and ENSO both have amplitudes of less than 0.05K at this height)

According to your suggestion, we have made a detailed analysis using the method in Randel and Wu (2014). Fig.2 shows the vertical profile of GNSS RO temperature variance in the deep tropics. The magnitude of annual cycle, QBO and ENSO related temperature anomalies shown in Fig. 2 is comparable to Randel and Wu (2014, Fig. 7). The residual at 150 hPa is much larger than the ENSO and QBO term at the same

level. This explains the residuals and the anomalies of the multivariate regression have same temporal structure and nearly the same magnitude. At 70 hPa the QBO50 term is much larger than ENSO and QBO30 terms but still less than the residuals.

[Figure]

Figure 3: Vertical profile of GNSS RO temperature variance in the deep tropics (10°S-10°N) associated with annual cycle, QBO, ENSO, and residual variability. The variance for the annual cycle has been divided by three to fit within this scale. The horizontal line denotes the altitude of the time average lapse rate tropopause.

6. Another concern I have with this study is that the connections between ozone and temperature are very loosely made, and there are no analyses to support them. Calculations (such as changes in temperature structure through changes in ozone through either a climate model or radiative transfer model) have not been made, and not even a simple correlation analysis was performed. Many previous studies (e.g. Abalos et al. 2012, Maycock 2016, Gilford et al. 2016, to name just a few) have done detailed modeling, radiative calculations or statistical analyses, quantifying the relationship between temperature and ozone. Instead, this paper simply notes "In the stratosphere, ozone distribution is highly correlated with the temperature change" (pg. 14, line 3) without actually showing any such correlations, and discusses some loose connections between temperature and ozone in section 3.4. Furthermore, it claims we need to "await further investigation" (pg. 3, line 27), but extensive research on this topic has been done! There is very little acknowledgement of the vast literature which has discussed this topic in detail, and the results herein are not framed within that context. Its important to perform some analysis to show how this work is valuable and contributing to our knowledge of ozone/temperature

links (especially in the context of how this relationship changes between reanalyses and GPS).

We apologize for didn't clearly introduce results about the connection between ozone and temperature in previous studies. A correlation analysis was performed between temperature anomalies and ozone anomalies from 2005 to 2017 and the potential contribution of ozone changes to temperature trends was also estimated. Fig.3 show the correlation coefficient between ozone and temperature and the ozone contributions to temperature trends. In general, all strong positive correlation (>0.6) between ozone and temperature can be found from 100 to 20 hPa. The correlation coefficients of ozone/T are highest in tropics (~0.9). The correlation coefficient between SWOOSH ozone and GNSS RO temperature is highest in average. MERRA2 shows a similar correlation between ozone and temperature while the correlation in ERA5 is slightly weaker. While ozone and temperature are positively correlated, a decrease of ozone contributes to a cooling in the NH and in the tropical upper troposphere and mid stratosphere. Increases of ozone lead to a warming effect in the SH and the lower stratosphere in the tropics.

[Figure]

Figure 4: The correlation coefficients between SWOOSH ozone and GNSS RO temperature (a), MERRA2 ozone/T (b) and ERA5 ozone/T (c), which are calculated from monthly deseasonalized anomaly time series from 2005 to 2017. The '+' marked the significant values using a p-value 0.05 for testing the hypothesis of no correlation. (d) SWOOSH ozone regressed GNSS RO temperature trends in K/decade; (e) MERRA2 ozone regressed temperature trends in K/decade; (f) ERA5 ozone regressed temperature trends in K/decade.

7. My primary concern with this paper is that it does not successfully and clearly distinguishing itself as novel. The trend calculations (for instance for ozone, pg. 3, line 21) have been updated through 2016 in previous studies, so this paper represents a2-year improvement (and as noted above, the depth ozone research herein is not at a level commensurate with previous studies). Studies of UTLS

temperature variability from GPS measurements have been very robustly presented in previous works (e.g. Abalos et al. 2012, Randel and Wu 2014). The use of the model to explore these processes is not well explained in the text, or compared with recently published studies which have done this (e.g. Randel et al. 2017).

To address this, I recommend the authors realign their motivation, highlighting that they are primarily concerned with comparing reanalyses and GPS in the UTLS with ERA5, in accordance with the S-RIP. Improvements in the ozone analyses and trend bias estimations in the context of comparing reanalyses will further improve on this narrative. Furthermore, the model should be brought introduced earlier in the paper as part of the motivation. This study can and will be valuable, but you need the tell and show the readers in clear language!

Thank you very much for the constructive comments. We agree to the reviewer that the motivation and the novel findings of this manuscript was not clearly addressed. We have rewritten the Introduction to highlight that our primary concern is to compare reanalysis data (in particular the ERA5 data) with the GPS-RO as the reviewer suggested. Other potential improvements of this manuscript than previous studies, i.e. an update of the temperature trend in the UTLS, the relationship between ozone and temperature changes and the attribution by model simulations, are also reorganized and addressed clearly in the revised manuscript.

Figure Comments:

All Figures: Please include units in all of your figure captions and titles/axes (where relevant).
Thank you for your remarks. We have added units in all figures.
Figure 1: One of the ranges in the caption should be "SM" instead of "NM". Also, it is not explained anywhere what is meant by SM and NM. Please add an explanation in the text of the manuscript.
Sorry for missing the information. The SM and NM indicate Southern hemisphere Mid-latitude and Northern hemisphere Mid-latitude, respectively. We have corrected the caption and added explanations in the revised manuscript.
Figures 4-5, 8-12, 14-15: Zonal mean figures would be improved if a line was added to indicate the climatological zonal mean tropopause height (using either the lapse rate tropopause or the cold-point tropopause, see Munchak and Pan 2014). These will likely vary from product to product and in the model, but it will help the read understand how your results vary with respect to the tropopause height.
Thank you for the suggestion. We have added the lapse rate tropopause in all figures.
Figures 1-3, 13: The x-axes on these timeseries plots are very hard to read because the years are all squished together.
Yes, we have renewed figures.
Figures 11-12, 14-15 (and timeseries plots): Readers who are green-red will find it

very difficult to parse the green "+" markers or green lines in these figures. Please use some other way or color contrast this data which is color-blind friendly.

Sorry, we have changed the green "+" markers to black.

Table 1: This is a key result in the entire paper, yet its unclear. What are the +/- values in this table, are they the confidence intervals from your t-test? If so, please indicate so. It's also important that trends in the biases from GPS RO be included as a column at each level, for comparison.

The +/- values in this table are 95% confidence intervals for the coefficient estimates. We have added the explanation in the text. The trends of the biases data are added in the table2.

Line-By-Line Comments:

Pg. 1, line 1: This first sentence is confusing as written.

We have rewritten this sentence.

Pg. 1, line 2 and elsewhere: Replace "were" with "are", and use present tense language throughout.

Thanks, we have checked carefully and updated the whole text.

Pg. 1, line 3+15: The first few sentences need to motivate the reader as to why your study is a valuable contribution and novel. I recommend mentioning the model here in addition to later, and be specific about what model you are using and in what mode.

Thank you for the kind suggestion. We have rewritten the sentences as suggested.

Pg. 1, line 13: replace "the change of" with "discontinuities in"

Corrected.

Pg. 1, line 16: The use of "could be" shows how the shallow the ozone and physically based analyses in this study are. Further analyses should allow you to be more definitive here.

Yes, we have changed it.

Pg. 2, line 1: It is not "the" key region, it is "a" key region. Coupling is also important at high latitudes (e.g. sudden stratospheric warmings).

Corrected.

Pg. 2, line 3: Do you mean that temperatures in the UTLS respond to climate change? That they affect other things (like water vapor) so they indirectly affect climate change? Please rewrite for clarity.

Yes, we have rewritten the sentence.

Pg. 2, lines 7-9: This sentence is confusing and should be rewritten.

Corrected.

Pg. 2, line 9: "through" should be "between"

Corrected.

Pg. 2, line 11: The term "underlying mechanisms" is used 4 times in this text without any clear explanation of what it means. Its use is vague and unspecific, please rewrite to clarify exactly what is meant when you say "underlying mechanisms".

"Underlying mechanisms" mean any possible mechanism/process that may influence the UTLS temperature, such as dynamical processes associated with SST, radiative effects by GHGs and ozone. We have updated the description in the manuscript.

Pg. 2, line 11: You are talking about trends in this paragraph, but now you mention variability (which could be construed as interannual variability). important to keep them distinct throughout the paper, because they could be changing in different ways.

Thank you for your suggestion and we have deleted the word.

Pg. 2, line 24: This is very poorly written sentence, please rewrite for clarity.

Corrected.

Pg. 2, line 27: "Plenty" is a slang term and not professional. Please look throughout your manuscript and replace these slang terms with more specific ones (e.g. "On one hand", pg. 3, line 4; "Same as", pg. 6, line 24; etc.). Here I suggest: "assimilate ground-based, satellite-based, and other data sources to provide the current..."

Thank you for your suggestion and we have corrected them in the text.

Pg. 2, line 31: The use of "perform" here is not correct. "may exhibit" would work. Other times in this paper "perform" is also not used correctly (e.g. pg. 13, line 24); please rewrite each of these.

Corrected.

Pg. 3, lines 1-2: This sentence is poorly written and distorts the communication of your goal.

We have rephrased this sentence.

Pg. 3, line 9: While ozone changes could be a helpful indicator as you claim, you've barely touched on how complicated this is. Schoeberl et al. (2008) did a rather complete study of this, but others (e.g. Polvani and Solomon 2012) have shown that it has rich nuances. You skip over that richness in your literature review here. I think its worth noting the efforts those papers made, and how your work is different.

Thank you for your suggestion and we have added literatures in the manuscript and the sentences to explain our work.

Pg. 3, line 10: "various of" should be "various"

Corrected.

Pg. 3, line 17: Very confusing as written.

Corrected.

Pg. 3, line 19: 15 hPa is well above the UTLS region!

We have deleted the sentence.

Pg. 3, line 29: The sentence is confusing as written.

Corrected.

Pg. 3, line 34: This a very abrupt transition introducing the model. This needs to be done more smoothly and with better motivation as to why we are using the model.

Yes, we have added one sentence before introducing the model.

Pg. 4, lines 3-10: Much of this paragraph is repetive with previous ones and can be removed.

Done.

Pg. 4, line 10: What is meant by "dynamical processing with SST"?

It means atmospheric circulation changes associated with SST. We have updated this sentence in the revised manuscript.

We thank the reviewer for all the comments and suggestions on the Introduction. The Introduction has been rewritten completely with all of comments considered.
Pg. 4, line 17: Seven years is not one decade. This is also very confusing as written.
Yes, we have changed the sentences.
Pg. 4, line 22: Are these measurement errors? Or differences from some other instrument?
They are estimated uncertainty for climate monitoring using GNSS radio occultation data.
Pg. 4, line 34: Can you provide a magnitude estimate for this "low effect"?
References show that less than 0.2K and I have added it in the manuscript.
Pg. 5, line 14: Was this linear interpolation done on a pressure grid or a height grid?
The linear interpolation has been done with logarithm pressure.
Pg. 5, line 17: What is meant by comparable here?
It means "similar".
Pg. 5, line 25: add "to" before "which"
Corrected.
Pg.5 line 27: There's no transition between these paragraphs. Are you introducing a new dataset you will also use?
Yes, we have added a sentence for transition as follows:
"For better study the ozone variability, an independent data sets namely C3S SAGE-II/CCI/OMPS ozone products version 3 with 10 ° latitude bands are used."
Pg. 6, line 2: On what basis can you call this "a time period suitable for trend evaluation"?
Sorry for the vague description. What we want to say here is that the C3S covers the year 2002 and 2017, which can be directly compared with SWOOSH data. We have corrected this sentence.
Pg. 6, line 7: introduce this as version 3 in the very first sentence of this paragraph instead.
We have introduced the version of data in the first sentence.
Pg. 6, line 16: As written, this sentence is unreadable. I don't understand what it is trying to say.
We have rewritten this sentence as follows:
"The newest ERA5 reanalysis, which is released by ECMWF in 2018, is also used."
Pg. 6, line 20: The link doesn't work as written, and should be more carefully cited in the bibliography.
Corrected.
Pg. 7, line 10: Please rewrite this confusing sentence.
We have rewritten this sentence as follows:
"The differences between these two simulations help to estimate the contribution of SST changes to temperature and ozone trends."
Pg. 7, line 11: I recommend renaming this section "Trend Calculations"

Updated.

Pg. 7, line 15: "Phenomenons" should be "phenomena"

Corrected.

Pg. 7, line 20: You have "a4" twice, but no solar component in equation 1.

Corrected.

Pg. 7, line 25: Is this a one-sided or two-sided t-test? Also, is this significance level the p-value? Please clarify your method.

It is a two-sided t-test and the significance level is 95%. We have clarified it in the text.

Pg. 7, line 29: The 400hPa level is well below the tropopause, especially in the tropics.

Thank you for your remarks. We use the Figure of 250hPa instead of the 400hPa level in the revised manuscript.

Pg. 8 line 11: What do you mean by "more disturbed" here?

The annual cycle at 100 hPa has substantial variability, which is not as regular as the annual cycle in the troposphere.

Pg. 9, line 22: why does the shortness of the period change this result? The shorter period means that interannual variability should have more influence on the trend calculations.

Yes, we have added the sentences in the text.

Pg. 9, line 27: "getting less" should be "smaller"

Corrected.

Pg. 9, line 29: The sentence is very confusing as written.

This sentence has been rewritten as follows:

"By such a multiple linear regression, the influences of ENSO and QBO as well as the linear trend can be separated."

Pg. 10, lines 4 and 12: What phase of ENSO or QBO? Please clarify throughout your paper what phase you mean each time you discuss results for QBO and ENSO.

Positive phase ENSO and westerly QBO. We have clarified the phase in the paper.

Pg. 10, line 17: This title isn't worded correctly. I suggest "Temperature Trends"

Corrected.

Pg. 10, line 28: I don't know what you mean by this sentence, you might be missing a word?

Corrected.

Pg. 10, line 31: "MEERA2" should be "MERRA2".

Corrected.

Pg. 11, line 5: Which tropopause? The cold point? The tropopause is a transition layer in the tropics (Fueglistaler et al. 2009)

The lapse rate tropopause.

Pg. 11, line 17: what dynamic process do you mean? Do you mean the influences of SSTs on circulation? If so, please say so.

Yes, we have changed it.

Pg. 11, line 28: "so many" should be "as many"

Corrected.

Pg. 12, line 35: This is a nice physical discussion which is mired by very unclear writing.

We have rewritten the discussion.

Pg. 13, line 1: Can you cite this? Many papers have shown this result.

Yes, we have cited previous studies.

Pg. 13, line 3: "That is not the truth" is not professional; please rewrite.

Yes, we have rewritten it.

Pg. 13, line 5: There is no observational evidence for ozone recovery yet, outside the spring SH stratosphere (Randel et al. 2017).

We have rewritten the sentence.

Pg. 13, line 16: You haven't done any attribution work, so this claim should be removed.

Corrected.

Pg. 13, line 22-24: These lines are very confusing; I don't understand what you mean.

We have updated the sentence as follows:

"ERA5 shows obvious improvements of temperature data compared with ERA-I and also a slight better agreement with GNSS RO measurements than MERRA2."

Pg. 13, line 29: 15 years is not "nearly 2 decades".

Corrected.

Pg. 14, line 1: This is a run-on sentence, and its very hard to parse what your point is here. Please rewrite.

This sentence has be updated as follows:

"Again, ERA5 shows improved quality compared with ERA-I and has the best agreement with the GNSS RO data in the three reanalyses."

Pg. 14, lines 3: You have not shown this result.

Yes, we have added the content.

Pg. 14, line 5: This result isn't true for all datasets in your study, and you haven't clarified what period these trends are considered over in this discussion.

We have clarified the period in the discussion.

Pg. 14, line 14: Your results do not show this link, please don't make false claims without evidence. In fact, it has been shown previously to not be the case (Randel et al. 2017).

We have deleted it.

Pg. 14, line 17: Poorly written.

Corrected.

References

Wargan, K., Orbe, C., Pawson, S., Ziemke, J. R., Oman, L. D., Olsen, M. A., Coy, L., Knowland, K. E: Recent decline in extratropical lower stratospheric ozone attributed to circulation changes. Geophysical Research Letters, 45, 5166–5176, https://doi.org/10.1029/2018GL077406, 2018

Wigley, T.: Appendix A: Statistical issues regarding trends, in: Temperature Trends in the Lower Atmosphere: Steps for Understanding and Reconciling Differences, edited by: Karl, T. R., Hassol, S. J., Miller, C. D., and Murray, W. L., A Report by Climate Change Science Program and the Subcommittee on Global Change Research, Washington, DC, USA, UNT Digital Library, 129–139, 2006.

---

## Author Comment (AC3) · 27 Mar 2019

**Short comments**

Dear Authors
Please consider these two comments, one scientific and one regarding data citation. This is an interesting paper and I hope you will find my remarks helpful.
Thank you very much for the very useful comments. We have updated the method and citation as suggested and hope the manuscript has been considerably improved.

Kind regards,

Ming Shangguan (on behalf of all co-authors)
1) I really appreciate your discussion of the negative impacts of step-changes in the ozone observing system on ozone trends in MERRA-2. Even a cursory look at Figure 13 reveals that the discontinuity associated with the transition from MLS v2.2 to v4.2 in June 2015 is nontrivial, as you correctly point out in Section 3.4. I would like to draw your attention to the fact that it is possible and relatively simple to account for this, as well as the 2004 SBUV-to-MLS transition, precisely because these step-changes are so infrequent and well defined. In Wargan et al, 2018 (doi:10.1029/2018GL077406) we did it using an SD model simulation as a transfer function but it could also be done by including a step-function proxy in the MLR. We tried the latter approach (not shown in our paper) and the result was very similar to that obtained using the transfer function approach. I suspect the MERRA-2 panel in Figure 14 would look different if a bias correction was applied. In fact, the analysis could be extended further back to 1998.
Thank you for your suggestion. We have corrected the discontinuity associated with the transition from MLS v2.2 to v4.2 in June 2015 with a step-function proxy in the MLR. The ozone trends have been updated in Figure 15 in the revised manuscript.

2) NASA GMAO asks the users of MERRA-2 data to explicitly cite the data collections used. Note that each MERRA-2 collection has a unique doi number listed in the file specs document https://gmao.gsfc.nasa.gov/pubs/docs/Bosilovich785.pdf For example monthly mean pressure-levels assimilated data ("M2IMNPASM" or *instM_3d_asm_Np*) could be cited as follows: Global Modeling and Assimilation Office (GMAO) (2015), MERRA-2 instM_3d_asm_Np: 3d,Monthly mean, Instantaneous, PressureLevel, Assimilation, Assimilated Meteorological Fields V5.12.4, Greenbelt, MD, USA, Goddard Earth Sciences Data and Information Services Center (GES DISC), Accessed: [Data Access Date], 10.5067/2E096JV59PK7
Thank you for the information. We have added the citation in the revised manuscript.

---

## Referee Report (RR1)

Second review of "Variability of temperature and ozone in the upper troposphere and lower stratosphere from multi-satellite observations and reanalysis data" by Shangguan et al. [Research Article acp-2018-1237]

Review Summary:
After revisions this paper is substantially improved, especially with regards to its motivation, literature review, and context. I appreciate the care the authors took to especially improve the introduction, and to add several analyses which provide more appropriate evidence for their conclusions; thank you. I am still concerned with some grammar issues in the text, but on that note the manuscript has also improved. With some additional editing to improve readability and a more careful write-up of the ozone results section, it is my conclusion that this manuscript will be fit for publication.

Major comment:

Figure 12: This figure is an excellent addition, but still needs a bit of work. The trends in the biases (i.e. trends in the differences between the reanalyses and GPS, akin to Randel and Wu 2006, Fig. 7) over the of the period of record should be included as an additional bar in this figure, for context, and discussed in the text. You essentially have this information in previous figures, but it is needed to provide context here to these trends.

Section 3.4: I continue to be concerned with the discussion in this section. It's clear from Figure 15 that there is essentially no agreement between either the datasets or the reanalyses (in contrast to what the authors claim on pg. 13, line 20). On this basis, it is very hard to judge what are the actual trends observed over this period, and it makes evaluation of them with the model (Fig. 16) less meaningful without knowing having confidence in the observational trend. Given the disagreement, it seems unlikely the trend analyses in Fig. 16 has a clear bearing on the real-world. The one clear consistent theme is that the tropical and SM lowermost stratosphere have increasing ozone trends and temperatures, while in the NM there is a clear decreasing trend in both temperature and ozone. This is mentioned briefly in conclusions (pg. 15 line 28), but it should be highlighted before this. The concluding paragraph in section 3.4 is good, but I still need to understand how it relates to the real-world, if the model and observations are not clearly in agreement.

Line-By-Line/Figure Comments:

Pg. 1 line 6: change "by" to "with"

Pg. 2 line 5: "SST increase" --> "warming SSTs"

Pg. 2 line 9: "greatly concerned" should be something like "extensively studied"

Pg. 2 line 15: Although I can tell what you mean, this line is confusing as written. May I suggest: "It is useful, therefore, to quantify the accuracy and variability of reanalysis temperature fields."

Pg. 2 line 25: add "it not being" before "susceptible"

Pg. 2 line 28: add "a" before "new generation"

Pg. 3 line 10: Rewrite as, "Long-term trends are a key issue in UTLS studies."

Pg. 4 line 11: The phrase "fully understand the exact reason" suggests that you have conclusively determined what the driving forces are, once and for all. This isn't a defendable point, as you note (and is clear from disagreements in literature) the interactions are very complex. I would suggest instead saying something like, "… are used in this study to investigate the reason".

Pg. 5 lines 15-17: You already say the range of your study in the preceding paragraph, so this is repetitive. You should either merge this with that line, or cut this line altogether.

Pg. 6 line 1: "For" should be "To"

Fig. 1: Is the weird spiking behavior of 2006 in NM at 250hPa related to a difference in when GPS observations were integrated into ERA5? It would be helpful to mention this (maybe at the top of pg. 9?) and explain it, because it stands out when you look at the differences figure.

Pg. 9 line 29: You should note that at higher altitudes in the lowermost stratosphere (70 hPa) ERA5 actually has the largest biases (~0.5K) in SM and NM, 2002-2006. It would be useful to write a line saying that no single reanalysis is universally better than the others, but (as you note in this line) on balance ERA5 appears to be best overall.

Pg. 11 line 7: This line isn't correct, and the nuance is important. Trends in the lowermost stratosphere are only significant in each of your datasets at 70hPa (and 50hPa additionally in the case of MERRA2). Please rewrite for accuracy.

Pg. 12 line 6: "specially concerned" --> "carefully considered"

Pg. 12, signal-to-noise: Thank you for including this analysis. It's much clearer and Fig. S2 makes it obvious to the reader that 16-year tropical trends are less meaningful. I would add a statement to this effect in line 16.

Pg. 12, lines 25-29: As rewritten and explained, this is a much stronger argument and more compelling evidence for your conclusions. Thank you!

Pg. 13 line 35: "dominated for" --> "dominate"

Pg. 13 lines 15-25: You need to note somewhere in here that the reanalysis and model trends look nothing like the observations… which casts doubt on the model's abilities to capture the behavior of the observations (and the relevance of the model results for the real world).

Pg. 13 line 35: In the 30-10hPa, I actually think that the fixed SST run is a much better representation of the observational datasets. It is important to be careful when describing this, as ozone and temperature trends should be expected to be increasingly linked as you go higher into the stratosphere.

---

## Author Response (AR2)

Second review of "Variability of temperature and ozone in the upper troposphere and lower stratosphere from multi-satellite observations and reanalysis data" by Shangguan et al. Research Article acp-2018-1237]

Review Summary:
After revisions this paper is substantially improved, especially with regards to its motivation, literature review, and context. I appreciate the care the authors took to especially improve the introduction, and to add several analyses which provide more appropriate evidence for their conclusions; thank you. I am still concerned with some grammar issues in the text, but on that note the manuscript has also improved. With some additional editing to improve readability and a more careful write-up of the ozone results section, it is my conclusion that this manuscript will be fit for publication.

We thank the reviewer for the positive feedbacks and the further comments. We have revised the manuscript accordingly based on the comments and suggestions. Please find details in the following point-to-point response as well as in the revised manuscript.

Major comment:
Figure 12: This figure is an excellent addition, but still needs a bit of work. The trends in the biases (i.e. trends in the differences between the reanalyses and GPS, akin to Randel and Wu 2006, Fig. 7) over the of the period of record should be included as an additional bar in this figure, for context, and discussed in the text. You essentially have this information in previous figures, but it is needed to provide context here to these trends.

We thank the reviewer for the useful comments. We have added the trends in the biases in Figure 12 as suggested.

Section 3.4: I continue to be concerned with the discussion in this section. It's clear from Figure 15 that there is essentially no agreement between either the datasets or the reanalyses (in contrast to what the authors claim on pg. 13, line 20). On this basis, it is very hard to judge what are the actual trends observed over this period, and it makes evaluation of them with the model (Fig. 16) less meaningful without knowing having confidence in the observational trend. Given the disagreement, it seems unlikely the trend analyses in Fig. 16 has a clear bearing on the real-world. The one clear consistent theme is that the tropical and SM lowermost stratosphere have increasing ozone trends and temperatures, while in the NM there is a clear decreasing trend in both temperature and ozone. This is mentioned briefly in conclusions (pg. 15 line 28), but it should be highlighted before this. The concluding paragraph in section 3.4 is good, but I still need to understand how it relates to the real-world, if the model and observations are not clearly

in agreement.

We agree to the reviewer that there are large disagreements between the two merged satellite data sets as well as reanalyses. We have rewritten the discussions regarding to the ozone trends and the coupling between ozone and temperature. Basically, we have pointed out clearly the large differences in ozone trends between different data sets. Then we focus on the consistent ozone trends in the two data sets, as mentioned by the reviewer, the positive ozone trends in the tropical and SM lowermost stratosphere and the negative ozone trends in the NM lower stratosphere (150-50 hPa) and mid stratosphere (30-10 hPa). The results from model simulations and the coupling between ozone and temperature is also discussed based on these consistent features.

Line-By-Line/Figure Comments:
Pg. 1 line 6: change "by" to "with"
Corrected.

Pg. 2 line 5: "SST increase" --> "warming SSTs"
It has been corrected.

Pg. 2 line 9: "greatly concerned" should be something like "extensively studied"
Done.

Pg. 2 line 15: Although I can tell what you mean, this line is confusing as written. May I suggest: "It is useful, therefore, to quantify the accuracy and variability of reanalysis temperature fields."
It has been changed.

Pg. 2 line 25: add "it not being" before "susceptible"
Corrected.

Pg. 2 line 28: add "a" before "new generation"
It has been added.

Pg. 3 line 10: Rewrite as, "Long-term trends are a key issue in UTLS studies."
Done.

Pg. 4 line 11: The phrase "fully understand the exact reason" suggests that you have conclusively determined what the driving forces are, once and for all. This isn't a defendable point, as you note (and is clear from disagreements in literature) the interactions are very complex. I would

suggest instead saying something like, "… are used in this study to investigate the reason".
It has been rewritten according to your suggestion.

Pg. 5 lines 15-17: You already say the range of your study in the preceding paragraph, so this is repetitive. You should either merge this with that line, or cut this line altogether.
The sentence has been deleted.

Pg. 6 line 1: "For" should be "To"
Corrected.

Fig. 1: Is the weird spiking behavior of 2006 in NM at 250hPa related to a difference in when GPS observations were integrated into ERA5? It would be helpful to mention this (maybe at the top of pg. 9?) and explain it, because it stands out when you look at the differences figure.

Yes, we have checked the start time of the COSMIC and it is exactly the time when the weird spiking behavior exists. We have added this discussion in the revised manuscript.

Pg. 9 line 29: You should note that at higher altitudes in the lowermost stratosphere (70 hPa) ERA5 actually has the largest biases (~0.5K) in SM and NM, 2002-2006. It would be useful to write a line saying that no single reanalysis is universally better than the others, but (as you note in this line) on balance ERA5 appears to be best overall.

We agree with the reviewer that ERA5 shows relative larger biases in SM and NM at 70 hPa. We have pointed out the relatively large bias of ERA5 at 70 hPa in the revised manuscript. However, if we compare the absolute values for the biases of different reanalyses (Figure 5), we still think ERA5 shows the best agreement with the GNSS RO, even at 70 hPa.

Pg. 11 line 7: This line isn't correct, and the nuance is important. Trends in the lowermost stratosphere are only significant in each of your datasets at 70hPa (and 50hPa additionally in the case of MERRA2). Please rewrite for accuracy.

Thanks for the comments. We agree with the reviewer that the trends at 50 hPa are not significant. We have addressed this clearly in the revised manuscript.

Pg. 12 line 6: "specially concerned" --> "carefully considered"
Done.

Pg. 12, signal-to-noise: Thank you for including this analysis. It's much clearer and Fig. S2 makes it obvious to the reader that 16-year tropical trends are less meaningful. I would add a statement to this effect in line 16. We have added a statement as suggested.

Pg. 12, lines 25-29: As rewritten and explained, this is a much stronger argument and more compelling evidence for your conclusions. Thank you!

Pg. 13 line 35: "dominated for" --> "dominate" Corrected.

Pg. 13 lines 15-25: You need to note somewhere in here that the reanalysis and model trends look nothing like the observations... which casts doubt on the model's abilities to capture the behavior of the observations (and the relevance of the model results for the real world).

Thanks for the comments. We agree with the reviewer that there are big differences between model and reanalysis and observations. We have modified the discussions and mainly focus on the regions with consistent ozone trends between different data sets (i.e. in the tropical lowermost stratosphere, the NM lower and mid stratosphere).

Pg. 13 line 35: In the 30-10hPa, I actually think that the fixed SST run is a much better representation of the observational datasets. It is important to be careful when describing this, as ozone and temperature trends should be expected to be increasingly linked as you go higher into the stratosphere.

We thank the reviewer for the comments. We have updated the discussions accordingly in the revised manuscript.

[revised manuscript text omitted]
  especially in the tropics. This indicate that large uncertainties exsit in the trends as shown in Figure 11 in the tropics. However, there are still significant signals in the mid-latitudes of the upper troposphere, around the tropopause and in the SH in the middle stratosphere. All the significant regions in Figure S2 are actually the most important areas with strongest and significant trends in Figure 11. This suggests that the significant trends shown in Figure 11 are robust except that in the tropics whereas the standard deviation of the trends are the strongest.

To explain the underlying mechanisms such as dynamical processes associated with SST of the illustrated temperature trends, two WACCM simulations as described in section 2.4 were employed. Figure 13 shows the temperature trends from the Transient run and the FixSST run as well as their differences. The Transient run with varying SST (Figure 13a) shows comparable positive trends (0.2-0.3 $K/decade$) in the troposphere and negative trends (0.1-0.5 $K/decade$) in the stratosphere (see Figure 11 for a comparison). While the SSTs are fixed to climatological values, which means only radiative effects from GHGs and ODSs are included, the positive trends in the troposphere disappear or become much weaker (Figure 13b). This reveals that the influences of SSTs on circulation are the main reason for the warming temperature trends in troposphere, which can be confirmed by the differences between these two runs (Figure 13c). The negative temperature trends in the stratosphere (tropics and SH) persist in the FixSST run, which illustrates other factors like radiative effects from GHGs and ozone contribute to such cooling. For the temperature trends above the tropical tropopause (100-50 hPa), the weak warming is related to combined effects of SSTs (contribute to a cooling) and other effects (lead to a warming).

**3.4 Coupling with ozone**

Figure 14 shows the initial ozone time series from the SWOOSH, C3S, MERRA2 and ERA5 as well as their differences using the SWOOSH data as a reference in three regions at 70 hPa. The ERA-I is not included here for ozone analysis because it does

not assimilate as many ozone measurements as ERA5 and MERRA2. Although the phase and amplitude agree well in general, the absolute ozone values have large differences between different data sets. Obvious missing data and extreme values exist in both SWOOSH and C3S data sets during 2002-2004, while a discontinuity in the MERRA2 and ERA5 time series occurs in mid-2004 when Aura MLS mission starts. As illustrated in Figure 14, extreme large values are observed by SWOOSH and
5  C3S around 2003. The reason is the limited number of observation in this period, which could cause large sampling errors and uncertainties in ozone data. At the same time, since the reanalysis is less constrained during this period, large bias can be seen in both MERRA2 and ERA5 compared to observations (Figures 14b, d and f). After 2006, SWOOSH uses MLS ozone data only (Davis et al., 2016) and MERRA2 also uses MLS instead of SBUV ozone data since Oct. 2004 (Gelaro et al., 2017). Therefore the MERRA2 ozone data have good agreement with SWOOSH data. Another discontinuity in the MERRA2 and
10  ERA5 time series occurs around 2015. According to McCarty et al. (2016), MERRA2 starts to use the version 4.2 MLS ozone data instead of version 2.2 since June 2015, which cause data discontinuities at 250-70 $\mathrm{hPa}$. As seen in Figures 14b, d and f, ozone in MERRA2 is 50-150 ppbv lower than that in SWOOSH and C3S. ERA5 combined more satellite data (SBUV and MLS) than MERRA2, which leads to larger variability of ozone in ERA5 since the different data sets and different ways for merging the data have large influences on the ozone data. The missing data and extreme values in SWOOSH and C3S, as well
15  as the data discontinuities in MERRA2 and ERA5 around years 2004 and 2015 can also be seen at other pressure levels (See Figures S3-S4 for details).

To examine the connection between the vertical temperature changes and ozone distribution, ozone trends are analyzed in the stratosphere from 250 to 10 $\mathrm{hPa}$. In consideration of the discontinuities in MERRA2 and ERA5 around late 2004 due to the MLS ozone data, a step-function proxy is added for the Jan. 2002-Sep. 2004 in the trend calculation. An extra step-function
20  proxy is added in the MERRA2 MLR to remove the discontinuities associated with the transition from MLS v2.2 to v4.2 for 250-70 $\mathrm{hPa}$ for the period Jun. 2015-Dec. 2017. The trends are calculated for the period 2002-2017 using the same MLR method as for temperature but with step function proxies in the reanalyses (Figure 15).  Large discepancies exist in the ozone trends between the two merged satellite data set (Figures 15a and b), which makes it hard to decide the actual trends of ozone for the period
25  2002-2017. This may be related to the large number of missing values in satellite observations in the early stage 2002-2004. While the trend are calculated from 2005 to 2017, the two data sets are more consistent to each other (not shown). Consistent negative trends in the NH lowermost stratosphere (150-50 hPa) and in the middle stratosphere (30-10 hPa) can be seen in both the SWOOSH and the C3S data sets, while the positive ozone trends in the tropical lower stratospere (100-50 hPa) are also in good agreement.
30   Asymmetry trends in two hemispheres, with significant decrease of ozone in NH mid-latitudes at 100-10 $\mathrm{hPa}$ and increase of ozone in SH mid-latitudes are found at  30-10 $\mathrm{hPa}$. This is consistent with a recent study using the MLS ozone data (Chipperfield et al., 2018).
35   Ozone trends in the reanalyses are different from the merged satellite data sets

as well as between each other. The only agreement can be seen in the positive trends of ozone in the lowermost stratosphere (150-50 hPa) in the tropics.

Figure 16 shows the ozone trends from two model simulations as well as their differences. The ozone trends based on the model simulation with varying SST show similar trends as SWOOSH and C3S data   for their consistent trends in the NH lowermost stratosphere (150-50 hPa), the NH mid stratosphere (30-10 hPa) and the tropical lower stratospere (100-50 hPa). While the SSTs are fixed to climatological values, ozone increases from the tropics to SH mid-latitudes in the middle stratosphere (30-10 hPa) and negative trends in the NH mid-latitudes from  30 to 10 hPa become much weaker (Figure 16b). The negative ozone trends in the NH lower stratosphere (150-50 hPa) seen in Figure 16a turn to the opposite in 
[revised manuscript text omitted]

[Figure]

**Figure 2.** As in Figure 1 but for 100 hPa.

(

[Figure]

**Figure 3.** As in Figure 1 but for 70 hPa.

[Figure]

**Figure 4.** The bias in temperature climatology as retrieved from CHAMP and COSMIC RO data. The two missions obtained data during a 28 month overlap period from Jun. 2006 to Sep. 2008. (a) The difference of monthly zonal mean temperature; (b) The corresponding averaged difference for each layer. The dash black lines marked the tropopause height calcualted with GNSS RO data.

[Figure]

**Figure 5.** Differences of temperature anomalies between three reanalyses and CHAMP from 400 to 10 hPa for 2002-2006 (a, c and e) and between three renalayses and COSMIC for 2007-2017 (b, d and f). The dash black lines marked the tropopause height calcualted with GNSS RO data.

[Figure]

**Figure 6.** Temperature anomaly at pressure level 150 hPa in the tropics(10°S-10°N) from ERA5 (green), ERA-I (light blue), MERRA2 (red) and GNSS RO (black) (a); (b) The corresponding residual; (c) The linear terms; (d) The QBO50 terms; (e) The QBO30 terms and (f) the ENSO terms; The solid lines in (c-f) marked the siginificant terms and the dash lines in (c-f) marked the insiginifcant terms.

[Figure]

**Figure 7.** As in Figure 6 but for 70 hPa.

[Figure]

**Figure 8.** ENSO related temperature anomalies of GNSS RO (a), ERA5 (b), MERRA2 (c) and ERA-I (d). The dash black lines marked the tropopause height calcualted with GNSS RO data.

[Figure]

**Figure 9.** As in Figure 8 but for QBO50.

[Figure]

**Figure 10.** As in Figure 8 but for QBO30.

[Figure]

**Figure 11.** Temperature trend in K/decade based on GNSS RO (a), ERA5 (b), MERRA2 (c) and ERA-I (d) data for period 2002-2017. The '+' marked the significant area at 95% level. The dash black lines marked the tropopause height calcualted with GNSS RO data.

[Figure]

**Figure 12.** Estimated temperature trends in K/decade during different regions (SM: 25°S-45°S; NM: 25°N-45°N; TP: 10°S-10°N) from 2002 to 2017. (a-f) Trends in corrected and uncorrected data sets at 250, 150, 70, 50, 20 and 10hPa. Trends based on the differences of time series between MEMRRA2/ERA5 and GNSS RO are also shown in the figure right. Error bars represent 95% confidence intervals.

[Figure]

**Figure 13.** Temperature trend in K/decade based on model simulations with time varying SST (a), fixSST (b) and their differences (c) for period 2002-2017. The '+' marked trends found to be more than 95% statistically significant. The dash black lines marked the tropopause height calcualted with GNSS RO data.

[Figure]

**Figure 14.** Left monthly mean ozone in ppbv at pressure level 70 hPa through three latitude bands of TP(10°S-10°N)(a), NM(25°N-45°N) (c), NM(25°S-45°S) (e); Right corresponding anomalies in figures (b), (d) and (e); Model with 103 levels (margin), ERA5 (green), C3S (light blue), MERRA2 (red) and SWOOSH (black) are included. The dash black lines marked the tropopause height calcualted with GNSS RO data.

[Figure]

**Figure 15.** Ozone trend in ppbv/decade based on SWOOSH (a), ERA5 (b), MERRA2 (c) and C3S (d) data for period 2002-2017. The '+' marked trends found to be more than 95% statistically significant. The dash black lines marked the tropopause height calculated with GNSS RO data.

[Figure]

**Figure 16.** Ozone trend in ppbv/decade based on model simulations with time varying SST (a), FixSST (b) and thier differences SST-fixSST (c) for period 2002-2017. (d) Model ozone related GNSS RO temeprature trends in K/decade. The '+' marked trends found to be more than 95% statistically significant. The dash black lines marked the tropopause height calcualted with GNSS RO data.